EMBO
Molecular Medicine

# Attenuation of clinical and immunological outcomes during SARS-CoV-2 infection by ivermectin

Guilherme Dias de Melo[1] , Françoise Lazarini[2] , Florence Larrous[1] , Lena Feige[1], Etienne Kornobis[3,4], Sylvain Levallois[5] , Agnès Marchio[6], Lauriane Kergoat[1] , David Hardy[7] , Thomas Cokelaer[3,4] , Pascal Pineau[6], Marc Lecuit[5,8] , Pierre-Marie Lledo[2], Jean-Pierre Changeux[9] & Hervé Bourhy[1,*]

## Abstract

The devastating pandemic due to SARS-CoV-2 and the emergence of antigenic variants that jeopardize the efficacy of current vaccines create an urgent need for a comprehensive understanding of the pathophysiology of COVID-19, including the contribution of inflammation to disease. It also warrants for the search of immunomodulatory drugs that could improve disease outcome. Here, we show that standard doses of ivermectin (IVM), an anti-parasitic drug with potential immunomodulatory activities through the cholinergic anti-inflammatory pathway, prevent clinical deterioration, reduce olfactory deficit, and limit the inflammation of the upper and lower respiratory tracts in SARS-CoV-2-infected hamsters. Whereas it has no effect on viral load in the airways of infected animals, transcriptomic analyses of infected lungs reveal that IVM dampens type I interferon responses and modulates several other inflammatory pathways. In particular, IVM dramatically reduces the *Il-6/Il-10* ratio in lung tissue and promotes macrophage M2 polarization, which might account for the more favorable clinical presentation of IVM-treated animals. Altogether, this study supports the use of immunomodulatory drugs such as IVM, to improve the clinical condition of SARS-CoV-2-infected patients.

**Keywords** coronavirus; inflammation; ivermectin; SARS-CoV-2; viral infections
**Subject Categories** Immunology; Microbiology, Virology & Host Pathogen Interaction

## Introduction

Coronaviruses cause respiratory disease in a wide variety of hosts. During the ongoing pandemic of SARS-CoV-2 causing coronavirus disease 19 (COVID-19), clinical signs other than respiratory symptoms have been linked to infection, frequently associated with neurological symptoms such as anosmia and ageusia. These features have been related to an over-responsiveness of patients' immune system to SARS-CoV-2 (Bhaskar *et al*, 2020; Han *et al*, 2020; Qiu *et al*, 2020). Consequently, there is an urgent need to understand the hallmarks of this over-responsiveness and to find novel therapeutics or repurpose drugs to improve the clinical condition of COVID-19 patients (Batalha *et al*, 2021).

Ivermectin (IVM), a macrocyclic lactone, is a commercially available anti-parasitic drug which prevents infection by a wide range of endo- and ectoparasites (Sajid *et al*, 2006; Heidary & Gharebaghi, 2020). IVM is an efficient positive allosteric modulator of the α-7 nicotinic acetylcholine receptor (nAChR) (Krause *et al*, 1998) and of several ligand-gated ion channels, including the muscle receptor for glutamate (GluCl) in worms (Hibbs & Gouaux, 2011). Furthermore, IVM has been shown to exert an immunomodulatory effect in humans and animals (Sajid *et al*, 2006; Heidary & Gharebaghi, 2020) under conditions that are known to involve the α-7 nAChR (Pavlov & Tracey, 2012), even though its underlying mechanisms are yet to be established (Laing *et al*, 2017). A direct or indirect interaction of SARS-CoV-2 with nAChR has also been hypothesized, in particular because of sequence homologies between SARS-CoV-2 spike proteins and nAChR ligands such as snake venom toxins (Changeux *et al*, 2020). IVM has been shown to be active beyond its anti-parasitic activity in a wide variety of pathologies, including cancer, allergy, and viral infections (Laing *et al*, 2017). Recently, IVM has been

1   Lyssavirus Epidemiology and Neuropathology Unit, Institut Pasteur, Paris, France
2   Perception and Memory Unit, Institut Pasteur, CNRS UMR 3571, Paris, France
3   Biomics Technological Platform, Center for Technological Resources and Research (C2RT), Institut Pasteur, Paris, France
4   Bioinformatics and Biostatistics Hub, Computational Biology Department, Institut Pasteur, Paris, France
5   Biology of Infection Unit, Institut Pasteur, Inserm U1117, Paris, France
6   Nuclear Organization and Oncogenesis Unit, Institut Pasteur, Paris, France
7   Experimental Neuropathology Unit, Institut Pasteur, Paris, France
8   Division of Infectious Diseases and Tropical Medicine, Institut Imagine, Université de Paris, Necker-Enfants Malades University Hospital, AP-HP, Paris, France
9   Neuroscience Department, Institut Pasteur, Collège de France, Paris, France
    *Corresponding author. Tel: +33 1 45 68 87 85; E-mail: herve.bourhy@pasteur.fr

reported to reduce viral load and improve the clinical status of mice infected by an animal coronavirus, the mouse hepatitis virus (MHV) (Arévalo *et al*, 2021). *In vitro* inhibition of SARS-CoV-2 replication by IVM in Vero/hSLAM cells has also been reported (Caly *et al*, 2020), albeit at much higher concentrations (50- to 100-fold) than those clinically attainable in humans (150–400 µg/kg) (Guzzo *et al*, 2002; Bray *et al*, 2020; Chaccour *et al*, 2020).

The aim of this study is to investigate the impact of IVM on the pathogenesis of COVID-19, in a SARS-CoV-2 infection model, the golden Syrian hamster. This species is naturally permissive to this virus and the most reliable and affordable animal model for COVID-19 (Chan *et al*, 2020; Muñoz-Fontela *et al*, 2020). Moreover, it was recently used to demonstrate the importance of lowering the inflammation with intranasal administration of type I IFN to prevent

**A** Clinical signs

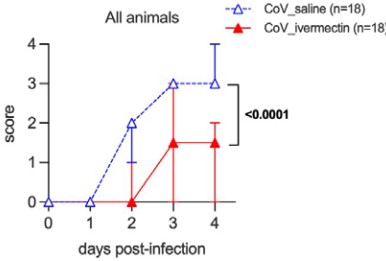
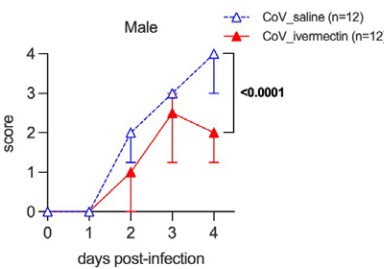
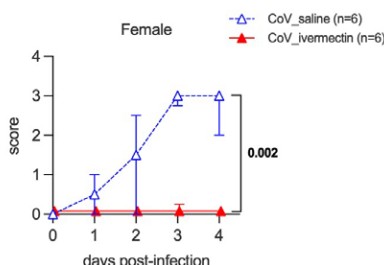

**B** Olfaction test at 3 dpi (food finding)

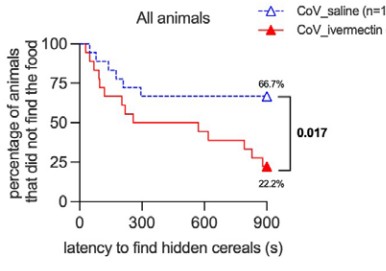
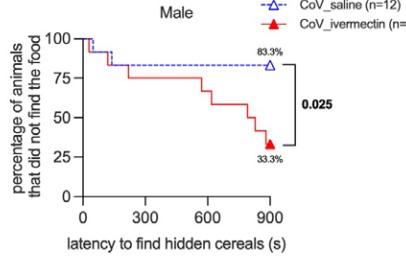
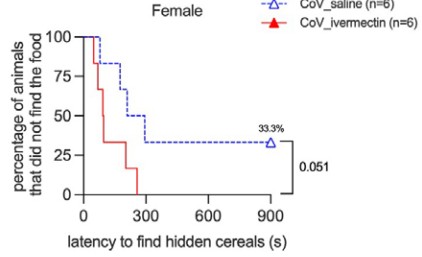

**C** Viral RNA load at 4 dpi

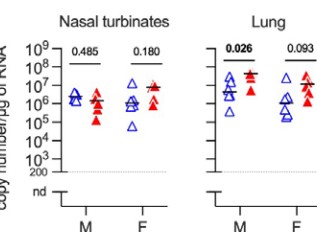

**D** ddPCR at 4 dpi

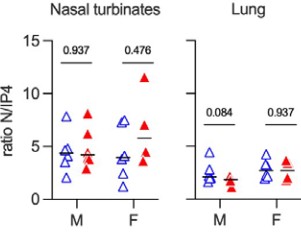

**E** Viral titer at 4 dpi

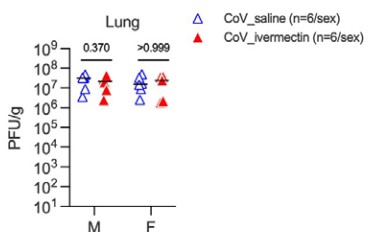

**F** Cytokines/chemokines in the nasal turbinates at 4 dpi

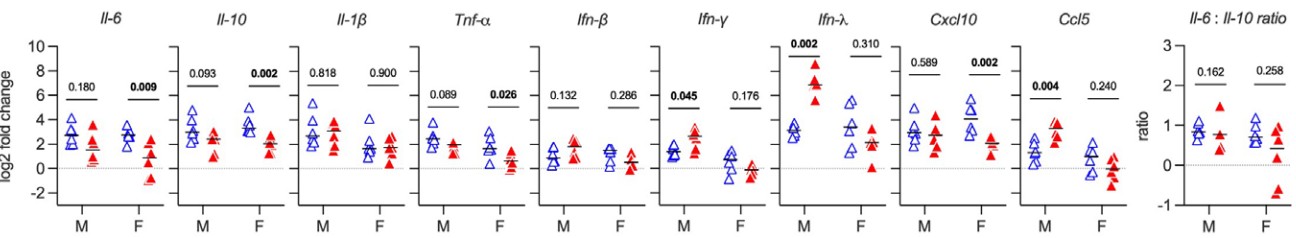

Figure 1.

**Figure 1.   Clinical presentation, olfaction test, viral load and immune profile in the nasal turbinates of SARS-CoV-2-infected hamsters with and without ivermectin treatment.**

A   Clinical signs in infected hamsters. The clinical score is based on a cumulative 0–4 scale: ruffled fur; slow movements; apathy; and absence of exploration activity. Symbols indicate the median ± interquartile range.

B   Olfactory performance in infected hamsters. The olfaction test is based on the hidden (buried) food finding test. Curves represent the percentage of animals that did not find the buried food. Food finding assays were performed at 3 days post-infection (dpi). Data were obtained from three independent experiments for males and two independent experiments for females.

C   Viral load in the nasal turbinates and in the lungs at 4 dpi.

D   Ratio between the CPD (copy per droplets, normalized to *γ-actin* and *Hprt* reference gene relative expression) of structural [N, nucleocapsid] and non-structural [IP4: RdRp, RNA-dependent RNA polymerase] viral gene expression determined by digital droplet PCR (ddPCR) in the nasal turbinates and in the lungs at 4 dpi.

E   Infectious viral titer in the lung at 4 dpi expressed as plaque-forming units (PFU)/g of tissue.

F   Cytokine and chemokine transcripts in the nasal turbinates at 4 dpi in male and female SARS-CoV-2-infected hamsters, treated with saline or with 400 μg/kg ivermectin.

Data information: Horizontal lines indicate medians. The *P* value is indicated in bold when significant at a 0.05 threshold. Mann–Whitney test (A, C–F) and log-rank (Mantel–Cox) test (B). M: male hamsters and F: female hamsters. Data were obtained from two independent experiments for each sex. See Figs EV1 and EV2 and Appendix Fig S1.

Source data are available online for this figure.

disease progression (Hoagland *et al*, 2020). Male and female adult golden Syrian hamsters were intranasally inoculated with $6 \times 10^4$ PFU of SARS-CoV-2 [BetaCoV/France/IDF00372/2020]. This inoculum size was selected as it invariably causes symptomatic infection in golden Syrian hamster, with a high incidence of anosmia and high viral loads in the upper and lower respiratory tracts within 4 days post-inoculation (dpi) (de Melo *et al*, 2021). At the time of infection, animals received a single subcutaneous injection of IVM at the anti-parasitic dose of 400 μg/kg, commonly used in human clinical setting, and were monitored over 4 days.

Here, we show that the modulation of the host's inflammatory response using IVM as a repurposed drug strongly diminished the clinical score and severity of the disease (including anosmia) observed in these animals, although it has no impact on viral load. IVM-treated animals presented a strong modulation in several signaling pathways, including a significant reduction of the type I and III interferon response and of the *Il-6/Il-10* ratio, along with the presence of M2 macrophages in the lung. These effects were mostly compartmentalized and sex-dependent, and treated infected females exhibited better clinical outcomes.

## Results and Discussion

### COVID-19 clinical outcome is attenuated by ivermectin

In order to study the effects of IVM chemical therapy on clinical outcome, we assessed body weight, clinical score, and olfactory performance daily for 4 days post-infection. The golden hamster model reproduces a moderate-to-severe COVID-19 in humans, with a clinical phase lasting 5–6 days post-infection followed by a complete recovery by 2 weeks post-infection, with no death occurring. The viral titers and viral RNA loads in the airways of infected animals are elevated after 2–4 days post-infection, but they drastically drop by 7 days post-infection (Chan *et al*, 2020; Sia *et al*, 2020; de Melo *et al*, 2021).

The IVM-treated infected animals, of both sexes, showed a decrease in body weight similar to that observed in saline-treated infected hamsters (Fig EV1A). However, IVM-treated infected animals exhibited a significant reduction in the severity of the clinical score, in a sex-dependent manner: Infected males presented a

significantly reduced clinical score, and it fully returned to normal in infected females (Fig 1A). Remarkably, IVM treatment reduced the olfactory deficit in infected animals (Figs 1B and EV1B–D): 66.7% (12/18) of the saline-treated infected hamsters presented with hyposmia/anosmia, whereas only 22.2% (4/18) of IVM-treated infected hamsters presented with signs of olfactory dysfunction (Figs 1B and EV1B–D). The olfactory performance was also influenced by sex: 83.3% (10/12) of the saline-treated infected males presented with hyposmia/anosmia, against only 33.3% (4/12) of IVM-treated infected males (Figs 1B and EV1B–D). In females, while 33.3% (2/6) of saline-treated infected animals presented with hyposmia/anosmia, no olfactory deficit was observed in IVM-treated infected females (0/6; Figs 1B and EV1B–D).

Since males presented a higher incidence of anosmia/hyposmia, we subsequently performed a dose-response curve to test the effect of IVM on the clinical presentation and olfactory functions of infected males: Lower doses of IVM (100 or 200 μg/kg) elicited similar clinical outcomes as the anti-parasitic dose of 400 μg/kg (Fig EV2). As expected, no signs of olfactory deficit were observed in mock-infected hamsters (Fig EV1B–D).

### Ivermectin treatment does not influence SARS-CoV-2 load in the respiratory tract of infected hamsters

To evaluate the effect of IVM treatment on the viral load in the respiratory tract, we tested the nasal turbinates and lungs of infected hamsters using both classical RT–qPCR (Fig 1C) and the highly sensitive technique of digital droplet PCR (Suo *et al*, 2020; Fig 1D, Appendix Fig S1). Surprisingly, the viral RNA load in the respiratory tract remained unaffected by IVM treatment in both samples in both sexes. Furthermore, IVM treatment did not influence viral replication rate, as evaluated by the ratio between structural and non-structural gene transcription (Fig 1D, Appendix Fig S1). Finally, IVM treatment did not alter infectious viral titers in the lungs (Fig 1E). These results illustrate that no antiviral activity of IVM is detected *in vivo* at the standard anti-parasitic dose of 400 μg/kg, in contrast to a previous report suggesting that IVM inhibits the replication of SARS-CoV-2 *in vitro*, albeit used at far higher concentrations (Caly *et al*, 2020). Therefore, the action of IVM on COVID-19 clinical signs of infected golden hamsters does not result from a decrease in viral replication.

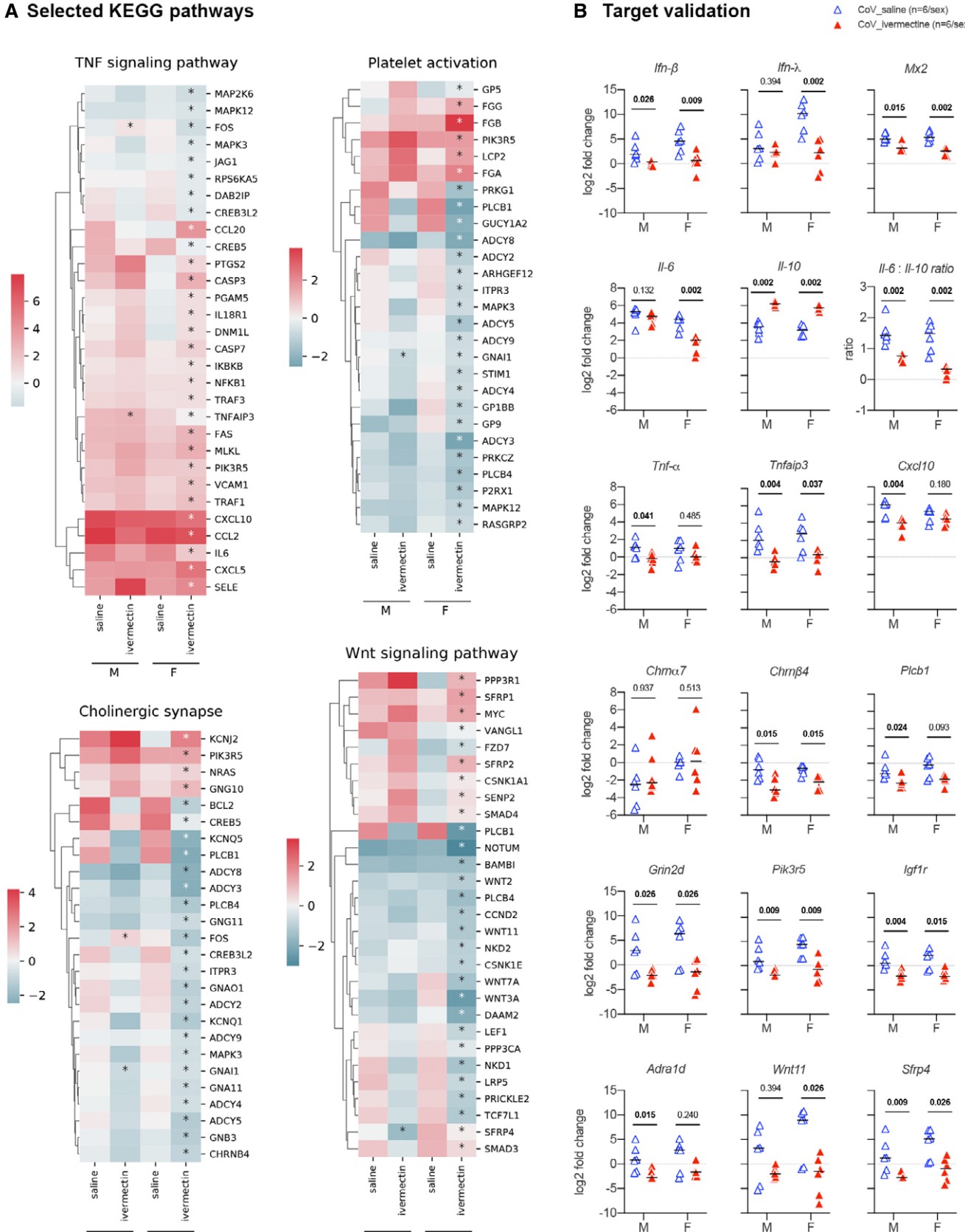

**Figure 2.**

◀

**Figure 2.  Transcriptomic profile in the lung of SARS-CoV-2-infected hamsters with and without ivermectin treatment at 4 days post-infection.**

A  Heatmaps showing the differentially expressed genes according to the selected KEGG pathways calculated in comparison with mock-infected hamsters. * indicates Benjamini–Hochberg-adjusted P-value < 0.05 in the comparison between saline and ivermectin within the same sex. Color gradient represents the transcription log$_2$ fold change comparing infected and mock-infected. Complete analyses are listed in Dataset EV1.
B  Validation targets in the lung at 4 dpi. Horizontal lines indicate medians. The P value is indicated in bold when significant at a 0.05 threshold. Mann–Whitney test.

Data information: M: male hamsters and F: female hamsters. Data were obtained from two independent experiments for each sex. See Figs EV3–EV5.
Source data are available online for this figure.

## Ivermectin therapy modulates local immune responses in infected hamsters' nasal turbinates

Anosmia is a typical symptom of COVID-19 in humans, with some sex-dependent differences (Han *et al*, 2020; Qiu *et al*, 2020; Xydakis *et al*, 2020). Inflammation in the nasal cavity, following olfactory sensory neurons infection and deciliation, has been shown to be an underlying factor for smell loss during SARS-CoV-2 infection (de Melo *et al*, 2021), and the chemokine *Cxcl10* could be directly implicated due to its neurotoxic potential (Oliviero *et al*, 2020). We therefore tested a possible modulation by IVM of the local inflammatory response in hamsters and in particularly in the nasal turbinates, the primary target tissue of SARS-CoV-2 infection (de Melo *et al*, 2021), that could correlate its effect on the olfactory score. To this aim, a panel of cytokines (*Il-6, Il-10, Il-1β, Tnf-α, Ifn-β, Ifn-γ,* and *Ifn-λ*) and chemokines (*Cxcl10* and *Ccl5*) were used to assess the impact of IVM treatment on immune responses in the nasal turbinates of SARS-CoV-2-infected hamsters at 4 dpi. Upon treatment with IVM, females presented a significant downregulation of *Il-6, Il-10, and Tnf-α*, which are key inflammatory mediators of prognostic value in COVID-19 patients (McElvaney *et al*, 2020a), and of *Cxcl10* (Fig 1F), in line with their better olfactory performance observed in the food finding tests (Fig 1B). The differences between sex groups are illustrated by the increase in three pro-inflammatory mediators (*Ifn-γ, Ifn-λ,* and *Ccl5*) only in males (Fig 1F). No difference for the *Il-6/Il-10* ratio was observed in the nasal turbinates.

## Lung immunometabolism is affected by SARS-CoV-2 infection and modulated by ivermectin

### IVM attenuates lung pathology and inflammation pathways, including cholinergic synapse-related genes

In order to further study the mode of action of IVM on clinical signs, we performed at 4 dpi a comparative agnostic transcriptomic approach using RNA-seq in the lower respiratory tract in hamsters treated or not with IVM. In SARS-CoV-2-infected lungs, male and female hamsters exhibited an overall similar pattern, although there were slightly more KEGG pathways modulated by infection in males (Fig 2A). A high number of common dysregulated inflammatory and metabolic pathways in SARS-CoV-2-infected males and females (Fig EV3) were similar to those observed in lung cells from human COVID-19 patients, such as "cytokine-cytokine receptor interaction", "TNF signaling pathway", and "insulin resistance" (Dey *et al*, 2020; Islam *et al*, 2020) including many genes of the IFN response. The transcriptomic profiles observed in the lungs from IVM-treated hamsters compared to non-treated infected hamsters presented a striking sex difference: Lungs of IVM-treated females presented 1,206 downregulated genes and 1,428 upregulated genes, whereas only 36 downregulated and 51 upregulated genes were detected in the lungs of IVM-treated males. This sex difference is also illustrated by KEGG and GO enrichments representations (Fig EV4).

Several KEGG pathways were significantly regulated in IVM-treated females: "TNF signaling pathway" and "cholinergic synapse", in line with the activation of vagus nerve cholinergic anti-inflammatory pathway (CAP) (Pavlov & Tracey, 2012; De Virgiliis & Di Giovanni, 2020), and "platelet activation", related to prevention of thrombosis (Zhang *et al*, 2020b; Chen & Pan, 2021) (Figs 2A and EV4A). We also observed a modulation of the "Wnt signaling pathway" KEGG pathway in the IVM treatment group, which has been shown to correlate with lung homeostasis and regeneration (Raslan & Yoon, 2020), obesity, type-2 diabetes, and cancer (Melotti *et al*, 2014; Aamir *et al*, 2020) (Figs 2A and EV4A). Further modulated pathways in IVM-treated female hamsters were "renin secretion", "vascular smooth muscle contraction", and "adrenergic signaling in cardiomyocytes" which could be related to the prevention of vasoconstriction and therefore attenuation of lung pathology (Fig EV4A) (Potus *et al*, 2020; Vaduganathan *et al*, 2020). Moreover, carbohydrate metabolism and insulin resistance were frequent terms observed in RNA-seq analyses of the lung of female hamsters during SARS-CoV-2 infection, linked to hyperglycemia, metabolic syndrome, and impairments in the immune system, similarly to what is observed in COVID-19 in humans (Pavlov & Tracey, 2012; Gianchandani *et al*, 2020). In IVM-treated hamsters, specifically in females, the term "insulin resistance" was no longer observed in the KEGG enrichment, and the insulin secretion pathway was downregulated. Several other related pathways linking hyperglycemia, inflammation, and lung pathology were downregulated, such as Rap1, AGE-RAGE, and cGMP-PKG signaling pathways (Figs EV4 and EV5) (Oczypok *et al*, 2017; Pei *et al*, 2020; Isidori *et al*, 2021). In IVM-treated males, significant modulated pathways determined by KEGG and GO enrichments were related to tight junctions, muscular and epithelial cells (Fig EV4).

Additionally, the GO enrichment in IVM-treated females revealed important modulated pathways including "regulation of neurotransmitter secretion" (Fig EV4B). Several important neurotransmitter receptors were indeed downregulated in both IVM-treated males and females, including cholinergic nicotinic (*Chrnb4*), adrenergic (*Adra1d*), GABAergic (*Gabbr1*), and glutamatergic (*Grin2d/ Nmdar2d, Grid1*), the latter possibly related to a protection of glutamate-induced lung injury (Said, 1999; Zhe *et al*, 2018).

### IVM exerts a strong regulation of type I and III IFN-related genes and lowers down the Il-6/Il-10 ratio in the lungs of SARS-CoV-2-infected hamsters

To analyze in more details the modulated genes in IVM-treated animals, we selected representative target genes from these relevant

## A Histopathological findings

## B Myeloid cells identification

## C Macrophages quantification

## D Macrophage polarization genes

## E Lung weight/body weight ratio

**Figure 3.**

**Figure 3. Identification of macrophages in the lung of SARS-CoV-2-infected hamsters with and without ivermectin treatment and characterization of their transcriptomic profile related to M1/M2 polarization.**

A  Representative histopathology photomicrographies of lungs according to the different groups: mock_saline, CoV_saline, and CoV_ivermectin. Top panels: whole lung sections. Bottom panels: high magnification. CoV_saline section exhibits important congestion (*), edema associated with few mononuclear cells (white arrowheads). Note the thickening of the alveolar walls. CoV_ivermectin section exhibits important amounts of mononuclear cells (black arrowheads) and less marked signs of congestion or edema. Hematoxylin and eosin. Scale bars = 1 mm (top panels) and 20 μm (bottom panels).
B  Representative immunofluorescence photomicrographies of neutrophils (Ly-6G), monocytes/macrophages (Iba1), M2 macrophages (Arg1), and SARS-CoV-2 (NP) in the lung. Scale bars = 50 μm.
C  Quantification of Iba1$^+$ cells, Arg1$^+$ cells, and Iba1$^+$Arg1$^+$ cells in the lungs. mock_saline $n = 3$ (males), mock_ivermectin $n = 4$ (males), CoV_saline $n = 9$ (6 males and 3 females), and CoV_ivermectin $n = 6$ (4 males and 2 females).
D  Heatmaps showing the differentially expressed genes related to the M1/M2 polarization in comparison with mock-infected hamsters. *indicates Benjamini–Hochberg-adjusted $P$-value < 0.05 in the comparison between saline and ivermectin within the same sex. Color gradient represents the transcription log$_2$ fold change comparing infected and mock-infected. Complete analyses are listed in Dataset EV1.
E  Lung weight-to-body weight ratio in the different groups ($n = 4$/sex/group).

Data information: M: male hamsters and F: female hamsters. Horizontal lines indicate medians. The $P$ value is indicated in bold when significant at a 0.05 threshold. Mann–Whitney test (C, E).
Source data are available online for this figure.

---

pathways that were highly regulated in IVM-treated females and we compared by RT–qPCR their respective transcription levels in the lungs of the different groups of animals. Among these genes, several were also significantly modulated between IVM- and saline-treated infected males including *Tnfaip3, Sfrp4, Epha2, Gnai1, Hgf,* and *Fos.* Others presented a similar regulation in males compared to females (from KEGG and GO enrichments) although their modulation was not invariably significant: *Casp3, Plcb1, Chrna7, Chrnb4, Adra1d, Grin2d/Nmdar2d, Grid1, Gabrr1, Pik3r5, Igf1r, Wnt11, Wnt3a, Il-2, Il-2ra, Prkg1, Krt4, Creb5,* and *Ager.* The modulation of these targets, together with other genes from relevant inflammatory mediators taken from the literature (Boudewijns *et al*, 2020; Hoagland *et al*, 2020) (*Il-6, Il-10, Il-1β, Tnf-α, Ifn-β, Ifn-λ, Ifn-γ, Tgf-β, Cxcl10, Ccl5,* and *Mx2),* was confirmed by RT–qPCR (Figs 2B and EV5).

IVM limited the expression of *Ifn-β* (males and females), *Ifn-λ* (females), and IFN-stimulated gene *Mx2* (males and females) in the lung of infected and IVM-treated hamsters compared to infected and saline-treated hamsters. In contrast, saline-treated hamsters presented an increased expression of *Ifn-β, Ifn-λ,* and *Mx2* compared to non-infected hamsters (males and females). This is expected as type I and III IFN signaling pathways have already been shown to correlate with lung pathology severity in SARS-CoV-2-infected hamsters, possibly resulting from a STAT2-dependent response (Boudewijns *et al*, 2020). In contrast, type I and III IFNs are differently expressed in the nasal turbinates, where *Ifn-λ* is upregulated in treated males (Fig 1F). This difference between upper and lower airways may be explained by two factors: (i) the specificity of *Ifn-λ* in the nasal cavity's antiviral response (Okabayashi *et al*, 2011; Klinkhammer *et al*, 2018), whereas type I and III IFNs perform redundant actions in the lung (Stanifer *et al*, 2020), and (ii), following studies with influenza A virus, type I and III IFNs have a sequential activation, where *Ifn-λ* comes at first to fight infection, and type I IFNs come later, with pro-inflammatory and reinforced antiviral effects (Galani *et al*, 2017; Stanifer *et al*, 2020).

Interestingly, some of these genes are not only involved in the previously mentioned pathways but also play a major role in other molecular functions that may be critical in the pathogenesis of SARS-CoV-2 infection and the efficacy of the IVM treatment in hamsters. Among them, the significant overexpression of *Il-10* was a common feature of IVM-treated males and females (Fig 2B). This

effect may be related to a modulation of the inflammatory response in the lung (downregulation of *Tnf-α* in males, *Tnfaip3* in both sexes, and *Il-6* in females) associated with reduced clinical signs. *Tnf-α* reduction and better clinical presentation are common features caused by the IVM treatment in another model of coronavirus infection, where mice were infected with MHV (Arévalo *et al*, 2021). Additionally, differently from the nasal turbinates, the *Il-6/Il-10* ratio in the lung of IVM-treated hamsters was significantly lower than in non-treated animals (Fig 2B), which may relate to their comparatively better clinical presentation. Of note, lower plasmatic *Il-6/Il-10* ratios were detected in hospitalized COVID-19 patients who did not require intensive care (McElvaney *et al*, 2020a; McElvaney *et al*, 2020b).

### IVM increases the infiltration of monocytes/macrophages and promote M2 polarization in the lung of SARS-CoV-2-infected hamsters

To assess directly lung pathology in infected animals, we performed histopathological analyses. The lungs of SARS-CoV-2-infected and saline-treated hamsters exhibited substantial lesions with focal areas of edema, congestion, microhemorrhages, associated with mononuclear cells, hyaline membranes, alveolar walls thickening (Fig 3A), in line with previous reports (Chan *et al*, 2020; Sia *et al*, 2020). In contrast, the lungs of SARS-CoV-2-infected and IVM-treated hamsters exhibited with reduced degrees of edema and congestion, yet with greater amounts of mononuclear cells in the alveolar spaces (Fig 3A). SARS-CoV-2 infection caused an increase in the lung weight/body weight ratio; however, this increase was lowered in IVM-treated as compared to saline-treated infected animals (Fig 3E).

The better clinical presentation observed in IVM-treated animals could be correlated with the activation of the cholinergic anti-inflammatory pathway (CAP) under vagus nerve control (Pavlov & Tracey, 2012). Indeed, the interaction of IVM with the α-7 nAChR in macrophages may account for the immune response modulation in IVM-treated hamsters (Gahring *et al*, 2017; Zhao *et al*, 2019), eliciting *Il-10* production, promoting M2 polarization (dampening inflammation and triggering tissue-repair), and counterbalancing M1 action (pro-inflammatory, associated with tissue damages and microbicidal activity) (Sang *et al*, 2015). This interpretation is supported by the upregulation of *Il-10* observed in the lung of IVM-

treated animals, along with the reduction of key M1 pro-inflammatory mediators, such as *Il-6, Tnf-α,* and *Cxcl10* (Fig 2B), in similar ways as observed in other viral infections (Sang *et al*, 2015).

Whereas few Iba1[+] cells were observed in mock-infected animals, most likely resident interstitial and alveolar macrophages, we observed a large number of Iba1[+] myeloid cells in the lungs of IVM-treated animals (Fig 3B). No distinguishable differences were observed in neutrophils population (Ly-6G[+]) between saline-treated and IVM-treated animals (Fig 3B). Part of these Iba1[+] cells were also Arg1[+] cells (Fig 3B), a marker of M2-polarized macrophages. IVM treatment was associated with an increase of both Iba1[+] and Arg1[+] cells in the lungs of SARS-CoV-2-infected male hamsters (Fig 3C). In contrast, in females, while infection promoted an increase of both Iba1[+] and Arg1[+] cells (Fig 3C), IVM had no impact on the recruitment of these cells. Additionally, RNA-seq analyses in the lung identified the upregulation of key M2-related genes (*Arg1, Cd209a/DC-SIGN, Clec7a/Dectin-1,* and *Myc/c-Myc*) along with classical M1 markers (*Cd86* and *Cd38*; Fig 3D), giving additional support to the M2 polarization tendency caused by the IVM treatment.

## Conclusions

Our results demonstrate that IVM improves clinical outcome in SARS-CoV-2-infected animals and is associated with a reduced inflammatory status, but with no impact of SARS-CoV-2 loads in the upper and lower respiratory tracts. Thus, in hamsters, as in humans (preprint: Cereda *et al*, 2020; Hasanoglu *et al*, 2021), symptomatology and therefore the severity of SARS-CoV-2 infection is not strictly correlated with viral load. The main effect of IVM in the lungs is on type I and III IFN responses and other related signaling pathways including phospholipases, kinases, and adenylate cyclases, which are important therapeutic targets (Melotti *et al*, 2014; Raker *et al*, 2016; Hu *et al*, 2020; Li *et al*, 2020; preprint: Masood *et al*, 2020; Isidori *et al*, 2021), and this translates clinically into an improved clinical score.

The results presented herein are consistent with a role of type I and III IFN responses in the pathogenesis of SARS-CoV-2-associated lung disease in hamsters. They show that IVM administration limits IFN response and lung inflammation, even though defects in the type I IFN pathways have been associated with severe COVID-19 (Bastard *et al*, 2020; Zhang *et al*, 2020a). This result may suggest that while IFN signaling is crucial to control viral replication and prevent severe disease, in infected hamsters, which only develop a moderate disease, IFN signaling may actually increase tissue damage and associated signs such as anosmia. This is consistent with previous reports in animal models where inhibition of IFN pathways was shown to increase viral replication but with lower lung pathology (Boudewijns *et al*, 2020; Sun *et al*, 2020).

Even if the effects observed may, to some extent, share similarities with the impact of dexamethasone (Horby *et al*, 2020) and tocilizumab (anti-IL-6) (Rossotti *et al*, 2020) on COVID-19, IVM action is steady and strong in the golden hamsters and is not expected to block the effectors of inflammation but rather to dampen its initiation. Interestingly, the activation of nuclear factor erythroid 2-related transcription factor (Nrf2) via low-dose radiotherapy (LDRT) treatment has been proposed to cause a shift from M1 to M2 macrophages and a blockade of NLRP3 inflammasomes (Calabrese *et al*,

2021) and to be potentially beneficial on the lungs of COVID-19 patients. Along the same line, we noticed via our RNA-seq analyses that IVM treatment increased the gene expression of Nrf2 (or Nfe2l2) in the lungs of female hamsters and slightly in males (Dataset EV1), which gives additional evidence of the broad and upstream activity of IVM during SARS-CoV-2 infection. Further, our data show that these effects are compartmentalized and that the upper and lower respiratory tracts of hamsters respond differently to IVM treatment.

Considerable sex differences are observed in terms of clinical presentation, inflammatory profile, and transcriptomic signatures in the lungs of hamsters, as seen in COVID-19 human patients, where men tend to develop more severe disease than women (Jin *et al*, 2020; Takahashi *et al*, 2020), possibly in relation with androgen signaling (vom Steeg & Klein, 2019; Samuel *et al*, 2020; Scully *et al*, 2020). Interestingly, sex steroids, here female hormones, might also influence both the course of COVID-19 in hamsters and the effects of IVM, possibly due to the potentiation of cell receptors signaling in females, such as nAChRs (Krause *et al*, 1998; Cross *et al*, 2017) and GlyRs (Van Den Eynden *et al*, 2009; Cerdan *et al*, 2020), of which IVM is a positive allosteric modulator.

Moreover, the data presented herein are consistent with those reported in human clinical trials with IVM (Kory *et al*, 2021). In humans, IVM is widely used as anti-helminthic and anti-scabies at therapeutic doses (150–400 µg/kg) (Guzzo *et al*, 2002) that are in the range of those used in our hamster experiments. Further, several clinical trials on COVID-19 using IVM have been declared, where IVM has been associated with reduction of inflammatory markers and disease severity (preprint: Hill *et al*, 2021; Kaur *et al*, 2021). Interestingly, in a long-term care facility where the residents received IVM to control a scabies outbreak, no death or severe COVID-19 was observed (Bernigaud *et al*, 2021). IVM has already been administered to hospitalized COVID-19 patients, with contrasting outcomes: one study related no efficacy of late IVM administration (8–18 days after symptom onset) in severe COVID-19 patients treated in combination with other drugs (hydroxychloroquine, azithromycin, tocilizumab, steroids) (Camprubí *et al*, 2020), whereas another study reported lower mortality, especially in severe COVID-19 patients treated with IVM in addition to other treatments (hydroxychloroquine, azithromycin, or both) (Rajter *et al*, 2020). Importantly, in a study that administered IVM alone within 72 h of symptom onset, the authors noticed an important diminution of anosmia/hyposmia in the IVM group without differences in PCR positivity between IVM and placebo groups (Chaccour *et al*, 2021).

The presently available data support the view that IVM pharmacokinetics is quite stable across species, generally with slow absorption, broad distribution, low metabolism, and slow excretion, even if some conditions, such as route of administration, formulation, and body condition, may modify these features. In humans, the oral route is the only approved, and the administration of a single dose of 30 mg of IVM (corresponding to 347–594 µg/kg) leads to a $C_{max}$ (maximum concentration) of 84.8 ng/ml, a $t_{max}$ (time to reach the $C_{max}$) of 4.3 h and a half-life of 20.1 h (Guzzo *et al*, 2002). In animal models, the subcutaneous route is most often used, and comparatively, golden hamsters that received a subcutaneous injection of 400 µg/kg of IVM (as in the present study) showed comparable results with a $C_{max}$ of 80.2 ng/ml and a $t_{max}$ of ~4 h (Hanafi *et al*, 2011). Further, based on veterinary data

(Lifschitz *et al*, 2000), simulations using a minimal physiologically based pharmacokinetic (mPBPK) model revealed that the lungs would be exposed to IVM concentrations 2.7× greater than those found in the plasma (Jermain *et al*, 2020). Yet, this dose did not suffice to achieve the range of antiviral concentrations reported *in vitro* (Caly *et al*, 2020).

Consequently, considering the results observed in the golden hamster model, IVM may be considered as a therapeutic agent against COVID-19, which would not strongly affect SARS-CoV-2 replication but limit the pathophysiological consequences of the infection *in vivo,* potentially mediated by type I and III IFN responses and several other related signaling pathways, and a favorable M1/M2 myeloid cells ratio in the lungs. A characteristic modulation of the immune response in the lower airways was observed in IVM-treated hamsters characterized by a transcriptomic profile similar to that observed in humans exhibiting less severe symptoms and a better prognosis (preprint: Masood *et al*, 2020; McElvaney *et al*, 2020b). Our data are consistent with the hypothesis that this effect is mediated by the cholinergic anti-inflammatory action of IVM on the vagus nerve reflex (Changeux *et al*, 2020; Tizabi *et al*, 2020), that should be addressed experimentally. In particular, the precise contribution of the nAChR in IVM action should be elucidated in comparison with that of other possible IVM targets (Zemkova *et al*, 2014). Altogether, this study brings the proof of concept that an IVM-based immunomodulatory therapy improves the clinical condition of SARS-CoV-2-infected hamsters, and in clinical trials, it alleviates symptoms of COVID-19 in humans and possibly limits post-COVID-19 syndrome (also known as long COVID) via an anti-inflammatory action.

# Materials and Methods

### Ethics

All animal experiments were performed according to the French legislation and in compliance with the European Communities Council Directives (2010/63/UE, French Law 2013–118, February 6, 2013) and according to the regulations of Pasteur Institute Animal Care Committees. The Animal Experimentation Ethics Committee (CETEA 89) of the Institut Pasteur approved this study (200023; APAFIS#25326-2020050617114340 v2) before experiments were initiated. Hamsters were housed by groups of 4 animals in isolators and manipulated in class III safety cabinets in the Pasteur Institute animal facilities accredited by the French Ministry of Agriculture for performing experiments on live rodents. All animals were handled in strict accordance with good animal practice.

### Production and titration of SARS-CoV-2 virus

The isolate BetaCoV/France/IDF00372/2020 (EVAg collection, Ref-SKU: 014V-03890) was kindly provided by Sylvie Van der Werf. Viral stocks were produced on Vero-E6 cells infected at a multiplicity of infection of $1 \times 10^{-4}$ plaque-forming units (PFU). The virus was harvested 3 days post-infection, clarified, and then aliquoted before storage at −80°C. Viral stocks were titrated on Vero-E6 cells by classical plaque assays using semisolid overlays (Avicel, RC581-NFDR080I, DuPont) (Baer & Kehn-Hall, 2014).

### SARS-CoV-2 model and ivermectin treatment of hamsters

Male and female Syrian hamsters (*Mesocricetus auratus;* RjHan: AURA) of 5–6 weeks of age (average weight 60–80 g) were purchased from Janvier Laboratories and handled under specific pathogen-free conditions. The animals were housed and manipulated in isolators in a Biosafety level-3 facility, with *ad libitum* access to water and food. Before manipulation, animals underwent an acclimation period of 1 week.

Animals were anesthetized with an intraperitoneal injection of 200 mg/kg ketamine (Imalgène 1000, Merial) and 10 mg/kg xylazine (Rompun, Bayer) and received one single subcutaneous injection of 200 μl of freshly diluted ivermectin (I8898, Sigma-Aldrich) at the classical anti-parasitic dose of 400 μg/kg (Beco *et al*, 2001) (or at 100–200 μg/kg for the dose–response experiment). Non-treated animals received one single subcutaneous injection of 200 μl of physiological solution. 100 μl of physiological solution containing $6 \times 10^4$ PFU of SARS-CoV-2 was then administered intranasally to each animal (50 μl/nostril). Mock-infected animals received the physiological solution only.

Infected and mock-infected animals were housed in separate isolators, and all hamsters were followed up daily during 4 days at which the body weight and the clinical score were noted. The clinical score was based on a cumulative 0–4 scale: ruffled fur, slow movements, apathy, and absence of exploration activity.

At day 3 post-infection (dpi), animals underwent a food finding test to assess olfaction as previously described (Lazarini *et al*, 2012; de Melo *et al*, 2021). Briefly, 24 h before testing, hamsters were fasted and then individually placed into a fresh cage (37 × 29 × 18 cm) with clean standard bedding for 10 min. Subsequently, hamsters were placed in another similar cage for 2 min when about five pieces of cereals were hidden in 1.5 cm bedding in a corner of the test cage. The tested hamsters were then placed in the opposite corner, and the latency to find the food (defined as the time to locate cereals and start digging) was recorded using a chronometer. The test was carried out during a 15-min period. As soon as food was uncovered, hamsters were removed from the cage. One minute later, hamsters performed the same test but with visible chocolate cereals, positioned upon the bedding. The tests were realized in isolators in a Biosafety level-3 facility that were specially equipped for that.

At 4 dpi, animals were euthanized with an excess of anesthetics (ketamine and xylazine) and exsanguination (AVMA, 2020), and samples of nasal turbinates and lungs were collected and immediately frozen at −80°C. Fragments of lungs were also collected and fixed in 10% neutral buffered formalin.

### RNA isolation and transcriptional analyses by quantitative PCR from golden hamsters' tissues

Frozen tissues were homogenized with TRIzol (15596026, Invitrogen) in Lysing Matrix D 2-ml tubes (116913100, MP Biomedicals) using the FastPrep-24™ system (MP Biomedicals) at the speed of 6.5 m/s during 1 min. Total RNA was extracted using the Direct-zol RNA MicroPrep Kit (R2062, Zymo Research: nasal turbinates) or MiniPrep Kit (R2052, Zymo Research: lung) and reverse-transcribed to first-strand cDNA using the SuperScript™ IV VILO™ Master Mix (11766050, Invitrogen). qPCR was performed in a final volume of 10 μl per reaction in 384-well PCR plates using a thermocycler

(QuantStudio 6 Flex, Applied Biosystems). Briefly, 2.5 µl of cDNA (12.5 ng) was added to 7.5 µl of a master mix containing 5 µl of Power SYBR Green Mix (4367659, Applied Biosystems) and 2.5 µl of nuclease-free water with nCoV_IP2 primers (nCoV_IP2-12669Fw: 5′-ATGAGCTTAGTCCTGTTG-3′; nCoV_IP2-12759Rv: 5′-CTCCCTTTGT TGTGTTGT-3′) at a final concentration of 1 µM (WHO, 2020). The amplification conditions were as follows: 95°C for 10 min, 45 cycles of 95°C for 15 s and 60°C for 1 min; followed by a melt curve, from 60 to 95°C. Viral load quantification of hamster tissues was assessed by linear regression using a standard curve of eight known quantities of plasmids containing the *RdRp* sequence (ranging from $10^7$ to $10^0$ copies). The threshold of detection was established as 200 viral copies/µg of RNA. The Golden hamster gene targets were selected for quantifying host inflammatory mediator transcripts in the tissues using the *Hprt* (hypoxanthine phosphoribosyltransferase), the *γ-actin,* and/or the actinB genes as reference (Appendix Table S1). Variations in gene expression were calculated as the *n*-fold change in expression in the tissues from the infected hamsters compared with the tissues of the uninfected ones using the $2^{-\Delta\Delta C_t}$ method (Pfaffl, 2001).

## Droplet digital PCR (ddPCR)

### Reverse transcription

200 ng of RNA was reverse-transcribed using iScript Advanced cDNA Synthesis kit for RT–qPCR (1702537, Bio-Rad) according to the manufacturer's specifications.

### Quantitative PCR for *γ*-actin *and Hprt reference genes*

Real-time PCR was performed in a CFX96 qPCR machine (Bio-Rad). All samples were measured in duplicate. The 10 µl PCR included 0.8 ng of cDNA, 1× PowerUp PCR master mix (A25742, Applied Biosystems), and 0.5 µM of each primer (Appendix Table S1). The reactions were incubated in a 96-well optical plate at 95°C for 2 min, followed by 40 cycles of 95°C for 15 s and 60°C for 1 min.

### Droplet digital PCR

ddPCRs were performed on the QX200 Droplet Digital PCR system according to the manufacturer's instructions (Bio-Rad). Briefly, reaction mixture consisted in 10 µl ddPCR Supermix for probe no dUTP (1863023, Bio-Rad), 0.25-1 ng of cDNA, primers and probes for E/IP4 and N/nsp13 duplex reactions used at concentration listed in Appendix Table S2 in a final volume of 20 µl. PCR amplification was conducted in a iCycler PCR instrument (Bio-Rad) with the following condition: 95°C for 10 min, 40 cycles of 94°C for 30 s with a ramping of 2°/s, 59°C for 1 min with a ramping of 2°/s, followed by 98°C for 5 min with a ramping of 2°/s and a hold at 4°C. After amplification, the 96-well plate was loaded onto the QX200 droplet reader (Bio-Rad) that measures automatically the fluorescence intensity in individual droplets. Generated data were subsequently analyzed with QuantaSoft™ software (Bio-Rad) based on positive and negative droplet populations. Data are expressed as CPD (copy per droplets) normalized to *γ-actin* and *Hprt* reference gene relative expression.

## Viral titration in golden hamsters' lung

Frozen lung fragments were weighted and homogenized with 1 ml of ice-cold DMEM supplemented with 1% penicillin/streptomycin (15140148, Thermo Fisher) in Lysing Matrix M 2-ml tubes (116923050-CF, MP Biomedicals) using the FastPrep-24™ system (MP Biomedicals) and the following scheme: homogenization at 4.0 m/s during 20 s, incubation at 4°C during 2 min, and new homogenization at 4.0 m/s during 20 s. The tubes were centrifuged at 10,000 *g* during 1 min at 4°C, and the supernatants were titrated on Vero-E6 cells by classical plaque assays using semisolid overlays (Avicel, RC581-NFDR080I, DuPont) (Baer & Kehn-Hall, 2014).

## Transcriptomics analysis in golden hamsters' lung

RNA preparation was used to construct strand-specific single-end cDNA libraries according to the manufacturers' instructions (TruSeq Stranded mRNA sample prep kit, Illumina). Illumina NextSeq 500 sequencer was used to sequence libraries. The RNA-seq analysis was performed with the Sequana framework (Cokelaer et al, 2017). We used the RNA-seq pipeline (v0.9.16), which is available online (https://github.com/sequana/sequana_rnaseq). It is built on top of Snakemake 5.8.1 (Köster & Rahmann, 2012). Reads were trimmed from adapters using Cutadapt 2.10 (Martin, 2011) and then mapped to the golden hamster MesAur1.0 genome assembly from Ensembl using STAR 2.7.3a (Dobin et al, 2012). FeatureCounts 2.0.0 (Liao et al, 2014) was used to produce the count matrix, assigning reads to features using annotation MesAur1.0.100 with strand-specificity information. Quality control statistics were summarized using MultiQC 1.8 (Ewels et al, 2016). Statistical analysis on the count matrix was performed to identify differentially regulated genes, comparing infected versus non-infected samples considering all samples and separating by sex. Clustering of transcriptomic profiles was assessed using a principal component analysis (PCA). Differential expression testing was conducted using DESeq2 library 1.24.0 (Love et al, 2014) scripts based on SARTools 1.7.0 (Varet et al, 2016) indicating the significance (Benjamini–Hochberg-adjusted *P*-values, false discovery rate FDR < 0.05) and the effect size (fold change) for each comparison. Finally, enrichment analysis was performed using modules from Sequana, first by converting golden hamster ensembl ids to gene names and then using human annotations for GO terms and KEGG pathways. The GO enrichment module uses PantherDB (Mi et al, 2019) and QuickGO (Huntley et al, 2014) services; the KEGG pathways enrichment uses gseapy (https://github.com/zqfang/GSEApy/), EnrichR (Chen et al, 2013), KEGG (Kanehisa & Goto, 2000), and BioMart services. All programmatic accesses to the online web services were performed via BioServices (Cokelaer et al, 2013).

## Histopathology

Lung fragments fixed in 10% neutral buffered formalin were embedded in paraffin. Four-µm-thick sections were cut and stained with hematoxylin and eosin staining. The slides were then scanned using Axioscan Z1 Zeiss slide scanner, using the Zen 2 blue edition software.

## Immunofluorescence

Lung fragments fixed in 10% neutral buffered formalin were washed in PBS and then embedded in O.C.T compound (4583, Tissue-Tek), frozen on dry ice, and cryostat-sectioned into 20-µm-thick sections. Sections were rinsed in PBS, and epitope retrieval was performed by

incubating sections for 20min in citrate buffer pH 6.0 (C-9999, Sigma-Aldrich) at 96°C for 20 min. Sections were then blocked in PBS supplemented with 10% goat serum, 4% fetal calf serum, and 0.4% Triton X-100 for 2 h at room temperature, followed by overnight incubation at 4°C with primary antibodies: rat anti-Ly6G (1/100, 551459, BD-Biosciences), chicken anti-Iba1 (1/500, 234006, Synaptic Systems), rabbit anti-Arg1 (1/250, PA5-29645, Invitrogen), and rabbit anti-SARS-CoV nucleoprotein (1/500, provided by Dr Nicolas Escriou, Institut Pasteur, Paris). After rinsing, slides were incubated with the appropriate secondary antibodies (1/500: goat anti-rat Alexa Fluor 546, A11081, Invitrogen; goat anti-rabbit Alexa Fluor 488, A11034, Invitrogen; goat anti-chicken Alexa Fluor 647, A32933, Invitrogen) for 2 h at room temperature. All sections were then counterstained with Hoechst (H3570, Invitrogen), rinsed thoroughly in PBS, and mounted in Fluoromount-G (15586276, Invitrogen) before observation with a Zeiss LM 710 inverted confocal microscope through a Plan Apochromat 20x/0.8 Ph2 M27 lens. Cell quantification was performed in an automated manner using ImageJ. Single-channel images were extracted, thresholded, and converted to binary images. Cells were then counted using the *Particles Analyzer* ImageJ plug-in.

## Statistics

Statistical analysis was performed using Prism software (GraphPad, version 9.0.0, San Diego, USA), with $P < 0.05$ considered significant. Quantitative data were compared across groups using log-rank test or two-tailed Mann–Whitney test. Randomization and blinding were not possible due to pre-defined housing conditions (separated isolators between infected and non-infected animals). *Ex vivo* analysis was blinded (coded samples). All animals were included, and data were provided from 2 replications, except food finding in males, that were replicated 3 times.

# Data availability

The datasets produced in this study are available in the following databases: RNA-seq: ArrayExpress E-MTAB-10128 (https://www.ebi.ac.uk/arrayexpress/experiments/E-MTAB-10128/).

Expanded View for this article is available online.

## Acknowledgements

The SARS-CoV-2 strain was supplied by the National Reference Centre for Respiratory Viruses hosted by Institut Pasteur (Paris, France) and headed by Dr. Sylvie van der Werf. The human sample from which strain 2019-nCoV/IDF0372/2020 was isolated has been provided by Dr. X. Lescure and Pr. Y. Yazdanpanah from the Bichat Hospital (Paris, France). This work was supported by Institut Pasteur TASK FORCE SARS COV2 (NicoSARS, NeuroCovid, and Cov-DROP projects) and received help from the European Union's Horizon 2020 Framework Programme for Research and Innovation under Specific Grant Agreement No. 945539 (Human Brain Project SGA3). We thank Elodie Turc and Laure Lemée, Biomics Platform, C2RT, Institut Pasteur, Paris, France, supported by France Génomique (ANR-10-INBS-09-09), IBISA and the Illumina COVID-19 Projects' offer. We would like to thank Marion Berard, Laetitia Breton, Rachid Chennouf, Hamidou Diakhate, and Eddie Maranghi for their help in implementing experiments in the Institut Pasteur animal facilities and Nicolas Escriou Innovation laboratory: Vaccines, Institut Pasteur, Paris, for providing the anti-SARS-CoV-2 nucleoprotein antibody. We thank Magali Tichit and Johan Bedel for the help with histopathology. We thank Gerard Orth, Arnaud Tarantola, and Andrew Holtz for critical reading of the manuscript. The synopsis illustration was created with BioRender.com.

## Author contributions

JPC and HB conceived the experimental hypothesis. GDM, FLaz, FLar, and HB designed the experiments. GDM, FLaz, FLar, LF, LK, SL, AM, and DH performed the experiments. GDM, FLaz, FLar, LF, EK, SL, AM, TC, PP, ML, and P-ML analyzed the data. GDM, J-PC, and HB wrote the manuscript, and all authors edited it.

## Conflict of interest

The authors declare that they have no conflict of interest.

## For more information

- COVID-19 section of the WHO website: https://covid19.who.int/
- COVID-19 section of the Institut Pasteur website: https://www.pasteur.fr/en/news-covid-19

# References

Aamir K, Khan HU, Sethi G, Hossain MA, Arya A (2020) Wnt signaling mediates TLR pathway and promote unrestrained adipogenesis and

## The paper explained

### Problem

The current pandemic of COVID-19 has caused more than 3.5 million deaths and more than 150 million laboratory-confirmed cases worldwide since December 2019 (as of May 2021). COVID-19, caused by SARS-CoV-2, commonly brings about upper airways and pulmonary symptoms and in severe cases can lead to respiratory distress and death. Different therapeutic approaches have been proposed to fight this disease but comprehensive therapeutic studies are still lacking.

### Results

We report that ivermectin, used at the standard anti-parasitic dose of 400 μg/kg, protects infected hamsters from developing clinical signs and from losing the sense of smell during SARS-CoV-2 infection. The treated animals exhibited a specific inflammatory response, presenting a reduced type I/III interferon stimulation and a modulation in several intracellular signaling pathways, with an important reduction of the *Il-6/Il-10* ratio and promoting M2 polarization of myeloid cells recruited to the lung. These effects are strongly influenced by sex, with treated females exhibiting the best outcome. Surprisingly, ivermectin treatment did not limit viral replication, as treated and non-treated animals presented similar amounts of SARS-CoV-2 in the nasal cavity and in the lungs.

### Impact

The results of this study establish that irrespective of viral load, the symptoms and severity of COVID-19 highlight the critical role played by host inflammatory response in COVID-19 severity and highlight that reduced type I/III interferon and *Il-6/Il-10* and the presence of M2 macrophages might account for a more favorable clinical presentation, contributing to a better understanding of COVID-19 pathophysiology. Ivermectin might then be considered as promising therapeutic agent against COVID-19 with no impact on SARS-CoV-2 replication but alleviating inflammation and ensuing symptoms.

metaflammation: therapeutic targets for obesity and type 2 diabetes. *Pharmacol Res* 152: 104602

Arévalo AP, Pagotto R, Pórfido Jl, Daghero H, Segovia M, Yamasaki K, Varela B, Hill M, Verdes JM, Duhalde Vega M *et al* (2021) Ivermectin reduces *in vivo* coronavirus infection in a mouse experimental model. *Sci Rep* 11: 7132

AVMA (2020) *AVMA guidelines for the euthanasia of animals: 2020 edition\**. Schaumburg, IL: American Veterinary Medical Association

Baer A, Kehn-Hall K (2014) Viral concentration determination through plaque assays: using traditional and novel overlay systems. *J Vis Exp* e52065

Bastard P, Rosen LB, Zhang Q, Michailidis E, Hoffmann HH, Zhang Y, Dorgham K, Philippot Q, Rosain J, Béziat V *et al* (2020) Autoantibodies against type I IFNs in patients with life-threatening COVID-19. *Science* 370: eabd4585

Batalha PN, Forezi LSM, Lima CGS, Pauli FP, Boechat FCS, de Souza MCBV, Cunha AC, Ferreira VF, da Silva FdC (2021) Drug repurposing for the treatment of COVID-19: Pharmacological aspects and synthetic approaches. *Bioorg Chem* 106: 104488

Beco L, Petite A, Olivry T (2001) Comparison of subcutaneous ivermectin and oral moxidectin for the treatment of notoedric acariasis in hamsters. *Veterinary Record* 149: 324–327

Bernigaud C, Guillemot D, Ahmed-Belkacem A, Grimaldi-Bensouda L, Lespine A, Berry F, Softic L, Chenost C, Do-Pham G, Giraudeau B *et al* (2021) Oral ivermectin for a scabies outbreak in a long-term–care facility: potential value in preventing COVID-19 and associated mortality? *Br J Dermatol* 184: 1207–1209

Bhaskar S, Sinha A, Banach M, Mittoo S, Weissert R, Kass JS, Rajagopal S, Pai AR, Kutty S (2020) Cytokine storm in COVID-19—immunopathological mechanisms. Clinical considerations, and therapeutic approaches: the REPROGRAM consortium position paper. *Front Immunol* 11: 1648

Boudewijns R, Thibaut HJ, Kaptein SJF, Li R, Vergote V, Seldeslachts L, Van Weyenbergh J, De Keyzer C, Bervoets L, Sharma S *et al* (2020) STAT2 signaling restricts viral dissemination but drives severe pneumonia in SARS-CoV-2 infected hamsters. *Nat Commun* 11: 5838

Bray M, Rayner C, Noël F, Jans D, Wagstaff K (2020) Ivermectin and COVID-19: a report in antiviral research, widespread interest, an FDA warning, two letters to the editor and the authors' responses. *Antiviral Res* 178: 104805

Calabrese EJ, Kozumbo WJ, Kapoor R, Dhawan G, Jimenez PCL, Giordano J (2021) NRF2 activation putatively mediates clinical benefits of low-dose radiotherapy in COVID-19 pneumonia and acute respiratory distress syndrome (ards): novel mechanistic considerations. *Radiother Oncol* 160: 125–131

Caly L, Druce JD, Catton MG, Jans DA, Wagstaff KM (2020) The FDA-approved drug ivermectin inhibits the replication of SARS-CoV-2 *in vitro*. *Antiviral Res* 178: 104787

Camprubí D, Almuedo-Riera A, Martí-Soler H, Soriano A, Hurtado JC, Subirà C, Grau-Pujol B, Krolewiecki A, Muñoz J (2020) Lack of efficacy of standard doses of ivermectin in severe COVID-19 patients. *PLoS One* 15: e0242184

Cerdan AH, Sisquellas M, Pereira G, Barreto Gomes DE, Changeux JP, Cecchini M (2020) The glycine receptor allosteric ligands library (GRALL). *Bioinformatics* 36: 3379–3384

Cereda D, Tirani M, Rovida F, Demicheli V, Ajelli M, Poletti P, Trentini F, Guzzetta G, Marziano V, Barone A *et al* (2020) The early phase of the COVID-19 outbreak in Lombardy, Italy. *arXiv* https://arxiv.org/abs/2003.09320 [PREPRINT]

Chaccour C, Hammann F, Ramón-García S, Rabinovich NR (2020) Ivermectin and COVID-19: keeping rigor in times of urgency. *Am J Trop Med Hyg* 102: 1156–1157

Chaccour C, Casellas A, Blanco-Di Matteo A, Pineda I, Fernandez-Montero A, Ruiz-Castillo P, Richardson M-A, Rodríguez-Mateos M, Jordán-Iborra C, Brew J *et al* (2021) The effect of early treatment with ivermectin on viral load, symptoms and humoral response in patients with non-severe COVID-19: a pilot, double-blind, placebo-controlled, randomized clinical trial. *EClinicalMedicine* 32: 100720

Chan JF, Zhang AJ, Yuan S, Poon VK, Chan CC, Lee AC, Chan WM, Fan Z, Tsoi HW, Wen L *et al* (2020) Simulation of the clinical and pathological manifestations of Coronavirus Disease 2019 (COVID-19) in golden Syrian hamster model: implications for disease pathogenesis and transmissibility. *Clin Infect Dis* 9: 2428–2446

Changeux J-P, Amoura Z, Rey FA, Miyara M (2020) A nicotinic hypothesis for Covid-19 with preventive and therapeutic implications. *CR Biol* 343: 33–39

Chen EY, Tan CM, Kou Y, Duan Q, Wang Z, Meirelles GV, Clark NR, Ma'ayan A (2013) Enrichr: interactive and collaborative HTML5 gene list enrichment analysis tool. *BMC Bioinformatics* 14: 128

Chen W, Pan JY (2021) Anatomical and pathological observation and analysis of SARS and COVID-19: microthrombosis is the main cause of death. *Biol Proced Online* 23: 4

Cokelaer T, Pultz D, Harder LM, Serra-Musach J, Saez-Rodriguez J (2013) BioServices: a common Python package to access biological Web Services programmatically. *Bioinformatics* 29: 3241–3242

Cokelaer T, Desvillechabrol D, Legendre R, Cardon M (2017) Sequana': a set of snakemake NGS pipelines. *J Open Source Softw* 2: 352

Cross SJ, Linker KE, Leslie FM (2017) Sex-dependent effects of nicotine on the developing brain. *J Neurosci Res* 95: 422–436

De Virgiliis F, Di Giovanni S (2020) Lung innervation in the eye of a cytokine storm: neuroimmune interactions and COVID-19. *Nat Rev Neurol* 16: 645–652

Dey A, Sen S, Maulik U (2020) Unveiling COVID-19-associated organ-specific cell types and cell-specific pathway cascade. *Brief Bioinform* 22: 914–923

Dobin A, Davis CA, Schlesinger F, Drenkow J, Zaleski C, Jha S, Batut P, Chaisson M, Gingeras TR (2012) STAR: ultrafast universal RNA-seq aligner. *Bioinformatics* 29: 15–21

Ewels P, Magnusson M, Lundin S, Käller M (2016) MultiQC: summarize analysis results for multiple tools and samples in a single report. *Bioinformatics* 32: 3047–3048

Gahring LC, Myers EJ, Dunn DM, Weiss RB, Rogers SW (2017) Nicotinic alpha 7 receptor expression and modulation of the lung epithelial response to lipopolysaccharide. *PLoS One* 12: e0175367

Galani IE, Triantafyllia V, Eleminiadou E-E, Koltsida O, Stavropoulos A, Manioudaki M, Thanos D, Doyle SE, Kotenko SV, Thanopoulou K *et al* (2017) Interferon-λ mediates non-redundant front-line antiviral protection against influenza virus infection without compromising host fitness. *Immunity* 46: 875–890.e6

Gianchandani R, Esfandiari NH, Ang L, Iyengar J, Knotts S, Choksi P, Pop-Busui R (2020) Managing hyperglycemia in the COVID-19 inflammatory storm. *Diabetes* 69: 2048–2053

Guzzo CA, Furtek CI, Porras AG, Chen C, Tipping R, Clineschmidt CM, Sciberras DG, Hsieh JY, Lasseter KC (2002) Safety, tolerability, and pharmacokinetics of escalating high doses of ivermectin in healthy adult subjects. *J Clin Pharmacol* 42: 1122–1133

Han AY, Mukdad L, Long JL, Lopez IA (2020) Anosmia in COVID-19: mechanisms and significance. *Chem Senses* 45: 423–428

Hanafi HA, Szumlas DE, Fryauff DJ, El-Hossary SS, Singer GA, Osman SG, Watany N, Furman BD, Hoel DF (2011) Effects of ivermectin on blood-feeding *Phlebotomus papatasi*, and the promastigote stage of *Leishmania major*. *Vector Borne Zoonotic Dis (Larchmont, NY)* 11: 43–52

Hasanoglu I, Korukluoglu G, Asilturk D, Cosgun Y, Kalem AK, Altas AB, Kayaaslan B, Eser F, Kuzucu EA, Guner R (2021) Higher viral loads in asymptomatic COVID-19 patients might be the invisible part of the iceberg. *Infection* 49: 117–126

Heidary F, Gharebaghi R (2020) Ivermectin: a systematic review from antiviral effects to COVID-19 complementary regimen. *J Antibiot* 73: 593–602

Hibbs RE, Gouaux E (2011) Principles of activation and permeation in an anion-selective Cys-loop receptor. *Nature* 474: 54–60

Hill A, Abdulamir A, Ahmed S, Asghar A, Babalola OE, Basri R, Chaccour C, Chachar AZK, Chowdhury ATM, Elgazzar A *et al* (2021) Meta-analysis of randomized trials of ivermectin to treat SARS-CoV-2 infection. *Res Squ* https://doi.org/10.21203/rs.3.rs-148845/v1 [PREPRINT]

Hoagland DA, Møller R, Uhl SA, Oishi K, Frere J, Golynker I, Horiuchi S, Panis M, Blanco-Melo D, Sachs D *et al* (2020) Leveraging the antiviral type-I interferon system as a first line defense against SARS-CoV-2 pathogenicity. *Immunity* 54: 557–570

Horby P, Lim WS, Emberson JR, Mafham M, Bell JL, Linsell L, Staplin N, Brightling C, Ustianowski A, Elmahi E *et al* (2020) Dexamethasone in hospitalized patients with Covid-19 - preliminary report. *N Engl J Med* 384: 693–704

Hu X, Cai X, Song X, Li C, Zhao J, Luo W, Zhang Q, Ekumi IO, He Z (2020) Possible SARS-coronavirus 2 inhibitor revealed by simulated molecular docking to viral main protease and host toll-like receptor. *Future Virol* 15: 359–368

Huntley RP, Sawford T, Mutowo-Meullenet P, Shypitsyna A, Bonilla C, Martin MJ, O'Donovan C (2014) The GOA database: gene ontology annotation updates for 2015. *Nucleic Acids Res* 43: D1057–D1063

Isidori AM, Giannetta E, Pofi R, Venneri MA, Gianfrilli D, Campolo F, Mastroianni CM, Lenzi A, d'Ettorre G (2021) Targeting the NO-cGMP-PDE5 pathway in COVID-19 infection. The DEDALO project. *Andrology* 9: 33–38

Islam T, Rahman MR, Aydin B, Beklen H, Arga KY, Shahjaman M (2020) Integrative transcriptomics analysis of lung epithelial cells and identification of repurposable drug candidates for COVID-19. *Eur J Pharmacol* 887: 173594

Jermain B, Hanafin PO, Cao Y, Lifschitz A, Lanusse C, Rao GG (2020) Development of a minimal physiologically-based pharmacokinetic model to simulate lung exposure in humans following oral administration of ivermectin for COVID-19 drug repurposing. *J Pharm Sci* 109: 3574–3578

Jin J-M, Bai P, He W, Wu F, Liu X-F, Han D-M, Liu S, Yang J-K (2020) Gender differences in patients with COVID-19: focus on severity and mortality. *Frontiers in Public Health* 8: 152

Kanehisa M, Goto S (2000) KEGG: kyoto encyclopedia of genes and genomes. *Nucleic Acids Res* 28: 27–30

Kaur H, Shekhar N, Sharma S, Sarma P, Prakash A, Medhi B (2021) Ivermectin as a potential drug for treatment of COVID-19: an in-sync review with clinical and computational attributes. *Pharmacol Rep* 73: 736–749

Klinkhammer J, Schnepf D, Ye L, Schwaderlapp M, Gad HH, Hartmann R, Garcin D, Mahlakõiv T, Staeheli P (2018) IFN-λ prevents influenza virus spread from the upper airways to the lungs and limits virus transmission. *eLife* 7: e33354

Kory P, Meduri GU, Varon J, Iglesias J, Marik PE (2021) Review of the emerging evidence demonstrating the efficacy of ivermectin in the prophylaxis and treatment of COVID-19. *Am J Ther* 28: e299–e318

Köster J, Rahmann S (2012) Snakemake—a scalable bioinformatics workflow engine. *Bioinformatics* 28: 2520–2522

Krause RM, Buisson B, Bertrand S, Corringer PJ, Galzi JL, Changeux JP, Bertrand D (1998) Ivermectin: a positive allosteric effector of the alpha7 neuronal nicotinic acetylcholine receptor. *Mol Pharmacol* 53: 283–294

Laing R, Gillan V, Devaney E (2017) Ivermectin – old drug, new tricks? *Trends Parasitol* 33: 463–472

Lazarini F, Gabellec M-M, Torquet N, Lledo P-M (2012) Early activation of microglia triggers long-lasting impairment of adult neurogenesis in the olfactory bulb. *The Journal of Neuroscience* 32: 3652–3664

Li N, Zhao L, Zhan X (2021) Quantitative proteomics reveals a broad-spectrum antiviral property of ivermectin, benefiting for COVID-19 treatment. *J Cell Physiol* 236: 2959–2975

Liao Y, Smyth GK, Shi W (2014) featureCounts: an efficient general purpose program for assigning sequence reads to genomic features. *Bioinformatics* 30: 923–930

Lifschitz A, Virkel G, Sallovitz J, Sutra JF, Galtier P, Alvinerie M, Lanusse C (2000) Comparative distribution of ivermectin and doramectin to parasite location tissues in cattle. *Vet Parasitol* 87: 327–338

Love MI, Huber W, Anders S (2014) Moderated estimation of fold change and dispersion for RNA-seq data with DESeq2. *Genome Biol* 15: 550

Martin M (2011) Cutadapt removes adapter sequences from high-throughput sequencing reads. *EMBnet J* 17: 10

Masood KI, Mahmood SF, Shahid S, Nasir N, Ghanchi N, Nasir A, Jamil B, Khanum I, Razzak S, Kanji A *et al* (2020) Transcriptomic profiling of disease severity in patients with COVID-19 reveals role of blood clotting and vasculature related genes. *medRxiv* https://doi.org/10.1101/2020.06.18.20132571 [PREPRINT]

McElvaney OJ, Hobbs BD, Qiao D, McElvaney OF, Moll M, McEvoy NL, Clarke J, O'Connor E, Walsh S, Cho MH *et al* (2020a) A linear prognostic score based on the ratio of interleukin-6 to interleukin-10 predicts outcomes in COVID-19. *EBioMedicine* 61: 103026

McElvaney OJ, McEvoy NL, McElvaney OF, Carroll TP, Murphy MP, Dunlea DM, Ní Choileáin O, Clarke J, O'Connor E, Hogan G *et al* (2020b) Characterization of the inflammatory response to severe COVID-19 illness. *Am J Respir Crit Care Med* 202: 812–821

de Melo GD, Lazarini F, Levallois S, Hautefort C, Michel V, Larrous F, Verillaud B, Aparicio C, Wagner S, Gheusi G *et al* (2021) COVID-19-related anosmia is associated with viral persistence and inflammation in human olfactory epithelium and brain infection in hamsters. *Sci Transl Med* 13: eabf8396

Melotti A, Mas C, Kuciak M, Lorente-Trigos A, Borges I, Ruiz i Altaba A (2014) The river blindness drug Ivermectin and related macrocyclic lactones inhibit WNT-TCF pathway responses in human cancer. *EMBO Mol Med* 6: 1263–1278

Mi H, Muruganujan A, Ebert D, Huang X, Thomas PD (2019) PANTHER version 14: more genomes, a new PANTHER GO-slim and improvements in enrichment analysis tools. *Nucleic Acids Res* 47: D419–D426

Muñoz-Fontela C, Dowling WE, Funnell SGP, Gsell P-S, Riveros-Balta AX, Albrecht RA, Andersen H, Baric RS, Carroll MW, Cavaleri M *et al* (2020) Animal models for COVID-19. *Nature* 586: 509–515

Oczypok EA, Perkins TN, Oury TD (2017) All the "RAGE" in lung disease: The receptor for advanced glycation endproducts (RAGE) is a major mediator of pulmonary inflammatory responses. *Paediatr Respir Rev* 23: 40–49

Okabayashi T, Kojima T, Masaki T, Yokota S-I, Imaizumi T, Tsutsumi H, Himi T, Fujii N, Sawada N (2011) Type-III interferon, not type-I, is the predominant interferon induced by respiratory viruses in nasal epithelial cells. *Virus Res* 160: 360–366

Oliviero A, de Castro F, Coperchini F, Chiovato L, Rotondi M (2020) COVID-19 pulmonary and olfactory dysfunctions: is the chemokine CXCL10 the common denominator? *The Neuroscientist* 27: 214–221

Pavlov VA, Tracey KJ (2012) The vagus nerve and the inflammatory reflex—linking immunity and metabolism. *Nat Rev Endocrinol* 8: 743–754

Pei J, Xiao Z, Guo Z, Pei Y, Wei S, Wu H, Wang D (2020) Sustained stimulation of β(2)AR inhibits insulin signaling in H9C2 cardiomyoblast cells through the PKA-dependent signaling pathway. *Diabetes Metab Syndr Obes* 13: 3887–3898

Pfaffl MW (2001) A new mathematical model for relative quantification in real-time RT-PCR. *Nucleic Acids Res* 29: e45

Potus F, Mai V, Lebret M, Malenfant S, Breton-Gagnon E, Lajoie AC, Boucherat O, Bonnet S, Provencher S (2020) Novel insights on the pulmonary vascular consequences of COVID-19. *Am J Physiol Lung Cell Mol Physiol* 319: L277–L288

Qiu C, Cui C, Hautefort C, Haehner A, Zhao J, Yao Qi, Zeng H, Nisenbaum EJ, Liu Li, Zhao Yu *et al* (2020) Olfactory and gustatory dysfunction as an early identifier of COVID-19 in adults and children: an international multicenter study. *Otolaryngol Head Neck Surg* 163: 714–721

Rajter JC, Sherman MS, Fatteh N, Vogel F, Sacks J, Rajter J-J (2020) Use of ivermectin is associated with lower mortality in hospitalized patients with COVID-19 (ICON study). *Chest* 159: 85–92

Raker VK, Becker C, Steinbrink K (2016) The cAMP pathway as therapeutic target in autoimmune and inflammatory diseases. *Front Immunol* 7: 123

Raslan AA, Yoon JK (2020) WNT signaling in lung repair and regeneration. *Mol Cells* 43: 774–783

Rossotti R, Travi G, Ughi N, Corradin M, Baiguera C, Fumagalli R, Bottiroli M, Mondino M, Merli M, Bellone A *et al* (2020) Safety and efficacy of anti-il6-receptor tocilizumab use in severe and critical patients affected by coronavirus disease 2019: A comparative analysis. *J Infect* 81: e11–e17

Said SI (1999) Glutamate receptors and asthmatic airway disease. *Trends Pharmacol Sci* 20: 132–135

Sajid MS, Iqbal Z, Muhammad G, Iqbal MU (2006) Immunomodulatory effect of various anti-parasitics: a review. *Parasitology* 132: 301–313

Samuel RM, Majd H, Richter MN, Ghazizadeh Z, Zekavat SM, Navickas A, Ramirez JT, Asgharian H, Simoneau CR, Bonser LR *et al* (2020) Androgen signaling regulates SARS-CoV-2 receptor levels and is associated with severe COVID-19 symptoms in men. *Cell Stem Cell* 27: 876–889.e12

Sang Y, Miller LC, Blecha F (2015) Macrophage polarization in virus-host interactions. *J Clin Cell Immunol* 6: 311

Scully EP, Haverfield J, Ursin RL, Tannenbaum C, Klein SL (2020) Considering how biological sex impacts immune responses and COVID-19 outcomes. *Nat Rev Immunol* 20: 442–447

Sia SF, Yan L-M, Chin AWH, Fung K, Choy K-T, Wong AYL, Kaewpreedee P, Perera RAPM, Poon LLM, Nicholls JM *et al* (2020) Pathogenesis and transmission of SARS-CoV-2 in golden hamsters. *Nature* 583: 834–838

Stanifer ML, Guo C, Doldan P, Boulant S (2020) Importance of type I and III interferons at respiratory and intestinal barrier surfaces. *Front Immunol* 11: 608645

vom Steeg LG, Klein SL (2019) Sex and sex steroids impact influenza pathogenesis across the life course. *Semin Immunopathol* 41: 189–194

Sun J, Zhuang Z, Zheng J, Li K, Wong R-Y, Liu D, Huang J, He J, Zhu A, Zhao J *et al* (2020) Generation of a broadly useful model for COVID-19 pathogenesis, vaccination, and treatment. *Cell* 182: 734–743.e5

Suo T, Liu X, Feng J, Guo M, Hu W, Guo D, Ullah H, Yang Y, Zhang Q, Wang X *et al* (2020) ddPCR: a more accurate tool for SARS-CoV-2 detection in low viral load specimens. *Emerg Microbes Infect* 9: 1259–1268

Takahashi T, Ellingson MK, Wong P, Israelow B, Lucas C, Klein J, Silva J, Mao T, Oh JE, Tokuyama M *et al* (2020) Sex differences in immune responses that underlie COVID-19 disease outcomes. *Nature* 588: 315–320

Tizabi Y, Getachew B, Copeland RL, Aschner M (2020) Nicotine and the nicotinic cholinergic system in COVID-19. *FEBS J* 287: 3656–3663

Vaduganathan M, Vardeny O, Michel T, McMurray JJV, Pfeffer MA, Solomon SD (2020) Renin–angiotensin–aldosterone system inhibitors in patients with covid-19. *N Engl J Med* 382: 1653–1659

Van Den Eynden J, SahebAli S, Horwood N, Carmans S, Brône B, Hellings N, Steels P, Harvey R, Rigo J-M (2009) Glycine and glycine receptor signalling in non-neuronal cells. *Front Mol Neurosci* 2: 9

Varet H, Brillet-Guéguen L, Coppée JY, Dillies MA (2016) SARTools: A DESeq2- and EdgeR-based R pipeline for comprehensive differential analysis of RNA-Seq data. *PLoS One* 11: e0157022

WHO (2020) *Protocol: real-time RT-PCR assays for the detection of SARS-CoV-2*, Paris: Institut Pasteur. https://www.who.int/docs/default-source/corona viruse/real-time-rt-pcr-assays-for-the-detection-of-sars-cov-2-institut-pa steur-paris.pdf?sfvrsn=3662fcb6_2

Xydakis MS, Dehgani-Mobaraki P, Holbrook EH, Geisthoff UW, Bauer C, Hautefort C, Herman P, Manley GT, Lyon DM, Hopkins C (2020) Smell and taste dysfunction in patients with COVID-19. *Lancet Infect Dis* 20: 1015–1016

Zemkova H, Tvrdonova V, Bhattacharya A, Jindrichova M (2014) Allosteric modulation of ligand gated ion channels by ivermectin. *Physiol Res* 63 (Suppl 1): S215–S224

Zhang Q, Bastard P, Liu Z, Le Pen J, Moncada-Velez M, Chen J, Ogishi M, Sabli IKD, Hodeib S, Korol C *et al* (2020a) Inborn errors of type I IFN immunity in patients with life-threatening COVID-19. *Science* 370: eabd4570

Zhang SI, Liu Y, Wang X, Yang LI, Li H, Wang Y, Liu M, Zhao X, Xie Y, Yang Y *et al* (2020b) SARS-CoV-2 binds platelet ACE2 to enhance thrombosis in COVID-19. *J Hematol Oncol* 13: 120

Zhao X, Yu Z, Lv Z, Meng L, Xu J, Yuan S, Fu Z (2019) Activation of alpha-7 nicotinic acetylcholine receptors (α7nAchR) promotes the protective autophagy in LPS-induced acute lung injury (ALI) *in vitro* and *in vivo*. *Inflammation* 42: 2236–2245

Zhe Z, Hongyuan B, Wenjuan Q, Peng W, Xiaowei L, Yan G (2018) Blockade of glutamate receptor ameliorates lipopolysaccharide-induced sepsis through regulation of neuropeptides. *Biosci Rep* 38: BSR20171629

