## [Review Process File · EMBO Molecular Medicine]

Attenuation of clinical and immunological outcomes during SARS-CoV-2 infection by ivermectin

Guilherme Melo, Françoise Lazarini, Florence Larrous, Lena Feige, Etienne Kornobis, Sylvain Levallois, Agnès Marchio, Lauriane Kergoat, David Hardy, Thomas Cokelaer, Pascal Pineau, Marc Lecuit, Pierre-Marie Lledo, Jean-Pierre Changeux, and Hervé Bourhy

DOI: [10.15252/emmm.202114122](https://doi.org/10.15252/emmm.202114122)

Corresponding author: Hervé Bourhy (herve.bourhy@pasteur.fr)

Review Timeline:

Submission Date:	11th Feb 21
Editorial Decision:	24th Mar 21
Revision Received:	8th Jun 21
Editorial Decision:	18th Jun 21
Revision Received:	22nd Jun 21
Accepted:	23rd Jun 21

Editor: Zeljko Durdevic

Transaction Report:

24th Mar 2021

Dear Dr. Bourhy,

Thank you for the submission of your manuscript to EMBO Molecular Medicine. We have now received feedback from the three reviewers who agreed to evaluate your manuscript. As you will see from the reports below, the referees acknowledge the interest of the study but also raise important criticism that should be addressed in a major revision.

Further consideration of the revision that addresses reviewers' concerns in full will entail a second round of review. EMBO Molecular Medicine encourages a single round of revision only and therefore, acceptance or rejection of the manuscript will depend on the completeness of your responses included in the next, final version of the manuscript. For this reason, and to save you from any frustrations in the end, I would strongly advise against returning an incomplete revision.

We would welcome the submission of a revised version within three months for further consideration. However, we realize that the current situation is exceptional on the account of the COVID-19/SARS-CoV-2 pandemic. Please let us know if you require longer to complete the revision.

I look forward to receiving your revised manuscript.

Yours sincerely,

Zeljko Durdevic

***** Reviewer's comments *****

Referee #1 (Remarks for Author):

EMM-2021-14122

This review is by Carlos Chaccour. ISGlobal - Barcelona Institute for Global Health. It is my usual practice to conduct unblinded reviews.

This is the report of study evaluating the clinical and immune effects of ivermectin on SARS-CoV-2 infection in golden hamsters. This is very timely work given an ongoing evaluation of ivermectin as a potential therapeutic for COVID-19 by the GHO Clinical Guideline Committee.

The experimental design of Dias de Melo et al. is very elegant and immunological evaluation is very thorough. The manuscript is well written, the methods are appropriate and the results support the conclusion.

I do have some technical comments worthy of discussion.

1. Adipose tissue, sex and ivermectin's PK

The authors have found a striking difference in the effects of the drug by animal sex which favors females. This has been partly explained by baseline differences in olfactory deficit (83% of males vs 33% of females present olfactory deficit in the control group) and it is suggested that androgen signaling could also play a role. In our own pilot trials cited by the authors (EClinicalMedicine 10.1016/j.eclinm.2020.100720), we also observe a difference in the response of anosmia to ivermectin, but in our case in favors the male participants.

The authors should consider discussing the pharmacokinetics of ivermectin as a potential contributing factor. Ivermectin is a highly lipophilic drug that binds eagerly to fat. Marked differences in its PK and systemic bioavailability have been described in relationship to body composition. Ouedraogo et al. (CID 2015) described higher ivermectin plasma concentrations in female participants in their clinical trial. A positive association between BMI and ivermectin plasma concentrations is also described. Both variables (sex and BMI) are intimately related to adipose weight (see Gómez-Ambrosi et al. Diabetes care 2012)

In contrast with humans and rats, in the golden Hamster it is the males that have higher adipose tissue (PMID: 12396289). Could the effect of ivermectin on the anosmia caused by SARS-CoV-2 be related to higher bioavailability of the drug (less fat-trapping) in individuals with less adipose weight i.e. male humans or female hamsters?

2. Drug-drug interactions

The Hamsters were anesthetized with a mixture of Ketamine-Xylazine.

Both anesthetic drugs are metabolized through the CYP 3A4 (the main metabolic pathway of ivermectin). This can increase the systemic exposure of drugs that undergo metabolism through this pathway, such as ivermectin (PMID 26489361). It is possible that drug-drug pharmacoenhancement may partly explain the efficacy observed at much lower doses.

Also, the partial inhibition of P-gp by both anesthetics may alter the distribution of the drug. This is important given the compartmental effect described by the authors.

There is a typo in olfactory in P5L21

Referee #2 (Comments on Novelty/Model System for Author):

The novelty of the approach is good but, is being applied by other researchers for both IVM and other small molecules that may be repurposed in such a way. The adequacy of the model system is good, for infectivity studies of SARS-CoV-2. However, the lack of hamster specific reagents does hinder experimentation/characterisation. The latter point is not due to the authors but, an issue in the field.

Referee #2 (Remarks for Author):

The authors present an interesting study on the immune modulatory, and potentially clinically beneficial, effects of ivermectin (IVM) in hamsters as a model of COVID-19 and infection with SARS-CoV-2. Following assessment of viral replication, in accordance with other studies, the authors show that IVM has no effect on viral replication in the model and further suggest that the potential benefits of IVM may lie in its immune modulatory abilities. To that end, the authors measured a number of immunological outcomes following infection with SARS-CoV-2 with and without IVM exposure. A number of interesting results are observed, specifically that there is a shift in the IL-6/IL-10 ratio, marked sex differences are observed in the responses to IVM, and a number of transcriptomic profile changes are apparent. The study is interesting, and warrants further investigation. However, prior to acceptance, there are a number of points that need to be addressed/clarified:

1. What is the exposure-response relationship in the animals? Doses are reported but not plasma/tissue concentrations; this is important for two reasons:
 - a. Accumulation within particular tissues, or cells, may be directly related to IVM effects
 - b. Differences in PK may, partially, explain discrepancies in RCT results in the literature
2. What explanation for the increased IFNs and CXCL5 in males but not females?
3. No impact on IL-1 β but inflammasome activation is apparent in COVID. Inflammasome activity is linked to severity of COVID-19 and NLRP3 activation has been reported as a response to viral proteins and cellular DAMPs. Targeting of inflammasome activation (Tranilast, pralnacasan) is under review in RCTs and modulation of inflammasome effects (Anakinra) has been shown to have favourable outcomes. How do the authors explain the lack of effects on these systems here and how it may result in favourable outcomes? Additionally, IVM has been shown to be a positive allosteric modulator of P2X7 and has been shown to exacerbate NLRP3 activation, yet no effect on IL-1 β was observed here.
4. A higher infiltration of Iba1 $^{+}$ monocytes/macrophages was observed in IVM treated animals but the authors did not conclude their activation status; as macrophage activation, in the lungs, is an important outcome in severity:
 - a. why did the authors not further define the profile of macrophages in the tissues, even if only by transcriptomics?
 - b. $\alpha 7$ nAChR activation increases HLA-DR, CD11b and CD54 expression but decreases CD14 (Siniavin, 2020) - how do the authors link to such findings?
5. Clinical manifestations are linked to immune mediated dysregulation of coagulation - any observations here? There were some actions on platelet pathways; how do the authors link that to possible beneficial outcomes?
6. What is the possible connection to nAChR here? It is mentioned several times but not expanded upon. It is known that activating $\alpha 7$ nAChR inhibits NLRP3 through β -arrestin-1 but no effect on IL-1 β was observed. Also, is there a differential expression profile of this receptor in immune cells? There are reported sex differences in B2 nAChR expression could that be true for $\alpha 7$ also?

Referee #3 (Comments on Novelty/Model System for Author):

Technical quality of the study is rated medium as more in-depth analyses would strengthen the manuscript (e.g. more detailed Analysis of cell recruitment to the respiratory tract). Novelty is high as treatments for COVID-19 are lacking and the study tests a novel candidate to modulate inflammation in COVID-19. Medical Impact is rated medium, as it is not clear how well data from the hamster model correlate to the patient's situation. Nevertheless the model system is adequate, as

permissive animal models are an important tool for analyses in vivo.

Referee #3 (Remarks for Author):

In their manuscript "Attenuation of clinical and immunological outcomes during SARS-CoV-2 infection by ivermectin" the authors analyze immune modulatory effects of ivermectin on SARS-CoV-2 induced disease in an experimental hamster model. Due to the immunological component in human COVID-19, the lack in effective treatments and the fact that ivermectin is an approved drug in humans, this is certainly an interesting and potentially relevant study. Overall, the manuscript is well written and the data are clearly presented. Nevertheless, there are some issues to be addressed to improve clarity and to strengthen the proposed therapeutic potential of ivermectin.

1) Introduction p 3, line 15:

The authors should provide more detailed information on how ivermectin modulates the host's immune system. Also it should be indicated whether these data are derived from human or animal studies. It does not become especially clear why the authors hypothesize an anti-parasitic drug to affect SARS-CoV-2 infection in a hamster model.

2) p 4, line 24:

Are the data referred to presented in Fig EV1 or rather Fig 1b? A p-value of 0.018 (Fisher's exact test) is stated in the text and a p-value of 0.017 (Mantel Cox) is stated in Fig 1b for the respective comparison. Please clarify how/why these statistic tests were performed.

"The olfactory performance was also influenced by sex: 83.3% (10/12) of the saline-treated infected males presented hyposmia/anosmia, against only 33.3% (4/12) of IVM-treated infected males (Fisher's exact test $p=0.036$). In contrast, no olfactory deficit was observed in IVM-treated infected females (0/6), while 33.3% (2/6) of saline treated infected females presented with hyposmia/anosmia (Fisher's exact test $p=0.455$)." When stating that olfactory performance is influenced by sex, shouldn't males and females (uninfected as well as infected) be compared directly and a p-value provided? Has olfaction been statistically analysed between uninfected and IVM-treated infected females? Looking at figure EV1b it seems that even though all IVM-treated infected females eventually find the food, they take longer to do so than the uninfected controls. Therefore, is it feasible to state that no olfactory deficit was observed in IVM-treated females (p4, line 26)?

3) p 7, line 23

"IVM limits the expression of *Ifn-b*, *Ifn-l* and of the IFN stimulated genes *Cxcl10* and *Mx2* in the lung of infected hamsters compared to infected and saline-treated hamsters."

As data are shown for females and males separately, the authors should indicate whether this statement refers to one or both sexes (as e.g. IFN- λ is not affected by IVM in males).

4) p4, line 24

"In contrast, saline treated hamsters presented an increased expression of *Ifn-l*, *Cxcl10* and *Mx2* compared to noninfected hamsters."

Please specify if this refers to males, females or both. For CXCL-10 also IVM treated hamsters still show about 4-fold higher transcription over uninfected hamsters. This should therefore be rewritten not to imply that expression is reduced to levels observed in uninfected hamsters.

5) p7, line 26

Regarding type I and III IFN, their induction following infection and IVM-mediated effects, the authors should discuss the differences observed between the nasal turbinates and the lung and what this

possibly implies for the observed overall host response/inflammation/clinical score. Generally, transcription of cytokine genes and effects of IVM in many points differ between the nasal turbinates and the lung and this should be discussed.

6) p7, line 34

Please also provide the IL-6/IL-10 ratio for the nasal turbinates. Is this a consistent finding between the lung and nasal turbinates?

7) p8, line 5

Recruitment of leucocytes to the lungs is an important aspect of the modulation of inflammation by IVM and the data shown are very interesting. However, more data should be presented to underline the findings. Only 2 mice were analyzed for the treatment group. For example, analysis of H&E stained lungs, differential cell counts of bronchoalveolar lavage or flow cytometric analyses could provide more insight into cell recruitment to the lungs in infection, effects of IVM and correlations to the cytokine/chemokine profile.

8) Regarding the timing of treatment in the model, what is the rationale for treating the hamsters together with the infection? Do the authors have any data on a more therapeutic potential of ivermectin, i.e. its effects when treating hamsters e.g. on day 2 or day 4 post infection when viral replication and infection have established?

9) The authors should provide more detailed information on the hamster model. Is it a lethal or sublethal model and is the virus eventually cleared. Here, hamsters were sacrificed 4 days post infection for analysis. To further assess the therapeutic potential, it would be important to assess whether IVM is able to improve survival. Along these lines, it is particularly stunning that IVM treated hamsters (especially females) show such little clinical scores until day 4 despite unchanged viral load in the respiratory tract. Also therefore, survival studies and a later time-point of analysis would need to confirm the potential of IVM. Possibly, in the later course the benefit of IVM treatment is overcome by a continuously high viral load.

10) Overall, from the data presented and discussed it does not become clear how the authors believe IVM acts upstream of the inflammatory response and how the strong difference between males and females regarding the effects of IVM treatment are mediated. Possibly the authors could elaborate more on these points.

Point-by-point answers to the Reviewers

Referee #1 (Remarks for Author):

EMM-2021-14122

This review is by Carlos Chaccour. ISGlobal - Barcelona Institute for Global Health. It is my usual practice to conduct unblinded reviews.

This is the report of study evaluating the clinical and immune effects of ivermectin on SARS-CoV-2 infection in golden hamsters. This is very timely work given an ongoing evaluation of ivermectin as a potential therapeutic for COVID-19 by the GHO Clinical Guideline Committee.

The experimental design of Dias de Melo et al. is very elegant and immunological evaluation is very thorough. The manuscript is well written, the methods are appropriate and the results support the conclusion.

I do have some technical comments worthy of discussion.

We wish to thank Carlos Chaccour for his positive comments. Below are our tentative answers to the different points he raises.

1. Adipose tissue, sex and ivermectin's PK

The authors have found a striking difference in the effects of the drug by animal sex which favors females. This has been partly explained by baseline differences in olfactory deficit (83% of males vs 33% of females present olfactory deficit in the control group) and it is suggested that androgen

signaling could also play a role. In our own pilot trials cited by the authors (EClinicalMedicine 10.1016/j.eclinm.2020.100720), we also observe a difference in the response of anosmia to ivermectin, but in our case in favors the male participants.

The authors should consider discussing the pharmacokinetics of ivermectin as a potential contributing factor. Ivermectin is a highly lipophilic drug that binds eagerly to fat. Marked differences in its PK and systemic bioavailability have been described in relationship to body composition. Ouedraogo et al. (CID 2015) described higher ivermectin plasma concentrations in female participants in their clinical trial. A positive association between BMI and ivermectin plasma concentrations is also described. Both variables (sex and BMI) are intimately related to adipose weight (see Gómez-Ambrosi et al. Diabetes care 2012)

In contrast with humans and rats, in the golden Hamster it is the males that have higher adipose tissue (PMID: 12396289). Could the effect of ivermectin on the anosmia caused by SARS-CoV-2 be related to higher bioavailability of the drug (less fat-trapping) in individuals with less adipose weight i.e. male humans or female hamsters?

R. It is known that the action of IVM might be affected by a variety of factors, including administration route, species and breed, age, body condition, including body fat. We did not evaluated adiposity in our study, therefore, we cannot assess that the male hamsters included in the study have more or less adipose tissue than the female hamsters.

Yet we first established that the body weights of males were similar to those of females.

Furthermore, we included in the discussion a section about IVM's pharmacokinetics in humans and hamsters as suggested by Carlos Chaccour, together with simulations regarding the lung exposure to the drug (page 11, lines 1-12). We could not include measures of IVM concentration in the plasma and lung of the hamsters specimens used in this study due to biosafety constraints. Indeed, we could not safely transfer the samples from the BSL-3 facilities to an external HPLC platform.

2. Drug-drug interactions

The Hamsters were anesthetized with a mixture of Ketamine-Xylazine.

Both anesthetic drugs are metabolized through the CYP 3A4 (the main metabolic pathway of ivermectin). This can increase the systemic exposure of drugs that undergo metabolism through this pathway, such as ivermectin (PMID 26489361). It is possible that drug-drug pharmaco-enhancement may partly explain the efficacy observed at much lower doses.

Also, the partial inhibition of P-gp by both anesthetics may alter the distribution of the drug. This is important given the compartmental effect described by the authors.

R. Ketamine-Xylazine anesthesia is a necessary component of animal experimentation. Its use is important to ensure animals' welfare and operator's safety. All animal manipulations in the context of COVID-19 research were performed with the use of these anesthetics (Chan et al., 2020, Imai et al., 2020, Kaptein et al., 2020, Sia et al., 2020). Drug-drug interactions might be a concern when testing the effects of a therapeutic protocol, but these questions need to be addressed by additional specific studies.

There is a typo in olfactory in P5L21

R. We corrected this in the new version.

Referee #2 (Comments on Novelty/Model System for Author):

The novelty of the approach is good but, is being applied by other researchers for both IVM and other small molecules that may be repurposed in such a way. The adequacy of the model system is good, for infectivity studies of SARS-CoV-2. However, the lack of hamster specific reagents does hinder experimentation/characterisation. The latter point is not due to the authors but, an issue in the field.

Referee #2 (Remarks for Author):

The authors present an interesting study on the immune modulatory, and potentially clinically beneficial, effects of ivermectin (IVM) in hamsters as a model of COVID-19 and infection with SARS-CoV-2. Following assessment of viral replication, in accordance with other studies, the authors show that IVM has no effect on viral replication in the model and further suggest that the, potential, benefits of IVM may lie in its immune modulatory abilities. To that end, the authors measured a number of immunological outcomes following infection with SARS-CoV-2 with and without IVM exposure. A number of interesting results are observed, specifically that there is a shift in the IL-6/IL-10 ratio, marked sex differences are observed in the responses to IVM, and a number of transcriptomic profiles changes are apparent. The study is interesting, and warrants further investigation. However, prior to acceptance, there are a number of points that need to be addressed/clarified:

1. What is the exposure-response relationship in the animals? Doses are reported but not plasma/tissue concentrations; this is important for two reasons:
 - a. Accumulation within particular tissues, or cells, may be directly related to IVM effects
 - b. Differences in PK may, partially, explain discrepancies in RCT results in the literature

R. We thank the reviewer for this important comment. Our serum, plasma and lung samples came from SARS-CoV-2-infected animals and were manipulated in dedicated BSL-3 facilities. Due to biosafety reasons, we are not able to provide these measurements in this revised version of the manuscript. Indeed, we could not safely transfer the samples from the BSL-3 facilities to an external HPLC platform. Considering the exceptional pandemics' situation and the long time needed to possibly be able to perform these analyses, we decided to submit the revised manuscript without these data, even if we keep working to find a solution to this issue.

Nevertheless, we have tried to provide this information to the readers from already published data, The pharmacokinetics of IVM is well documented in humans (González Canga et al., 2008, Guzzo et al., 2002) and in several animal species, including hamsters (González Canga et al., 2009, Lifschitz et al., 2000) and we added in the discussion a section about IVM's pharmacokinetics in humans and hamsters, including simulations about the lung exposure to IVM (page 11, lines 1-12).

2. What explanation for the increased IFNs and CCL5 in males but not females?

R. The sex-dependence of the immune response is an important issue that deserves specific studies, which are far beyond the aim of this paper. We do not have, at this stage, a definitive answer to this reviewer's question, but some hypotheses might be suggested based on the observed sex dependence of the immune response in other systems. It is known that, using other viruses such as influenza A virus in a model of respiratory infection, female mice exhibit a more severe inflammatory status than males, despite similar viral loads (vom Steeg & Klein, 2019). In the same line, sex-dependent outcomes are observed in COVID-19 patients (Scully et al., 2020, Takahashi et al., 2020) and type I IFNs responses as well as B-cell and T-cell functions have been reported as being sex-dependent (Bunders & Altfeld, 2020). The effects of IVM may also be potentiated by female steroids, such as 17 β -Estradiol, by modulation of cell receptors such as nAChRs and GlyRs (Cerdan et al., 2020, Cross et al., 2017). These possibilities are mentioned in page 10 (lines 14-18) of the revised version.

3. No impact on IL-1b but inflammasome activation is apparent in COVID. Inflammasome activity is linked to severity of COVID-19 and NLRP3 activation has been reported as a response to viral proteins and cellular DAMPs. Targeting of inflammasome activation (Tranilast, pralnacasan) is under review in RCTs and modulation of inflammasome effects (Anakinra) has been shown to have favourable outcomes. How do the authors explain the lack of effects on these systems here and how it may result in favourable outcomes? Additionally, IVM has been shown to be a positive allosteric modulator of P2X7 and has been shown to exacerbate NLRP3 activation, yet no effect on IL-1b was observed here.

R. Modulation of inflammasome activity is a rising field of research in anti-COVID-19 therapeutics. Therefore, we thank the reviewer for this excellent comment that will help the reader to understand more about the potential explanations regarding the role of IVM in SARS-CoV-2-infected hamsters. Our RNAseq analyses did not reveal differences in the gene expression of NLRP3 between the lungs of infected and non-infected animals. Further, no differences were found between the lungs of infected animals, treated or not by IVM (Dataset EV1, new Fig 3d). The same holds for IL-1 β . IL-1 β does not seem to play an extensive role in the immune response in the lungs, neither in IVM-treated nor in saline-treated hamsters (Fig EV5b). Therefore, our data are consistent with the view that there is no effect of IVM on the NLRP3/IL-1 β pathway. However, the new Fig 3d illustrates that many other genes involved in macrophages polarization (M1 x M2) are modulated by the IVM treatment. This is now discussed in page 8 (lines 28-34) and page 9 (lines 1-14).

P2X7 is known to be involved in the release of proinflammatory chemokines CCL-3, CCL-4 and CCL-5 and in the recruitment of CD8 T lymphocytes, however none of these effects are reported here (see Fig EV5b and not shown for CD8 T lymphocytes). P2X7 purinergic receptors modulated by IVM might plausibly be present in hamsters, but according to our preliminary non-published observations, the main anti-COVID-19 action of IVM is mediated by the nicotinic receptor. Studies are currently being performed to further refine this pharmacological specificity (see point 6 below).

4. A higher infiltration of Iba1+ monocytes/macrophages was observed in IVM treated animals but the authors did not conclude their activation status; as macrophage activation, in the lungs, is an important outcome in severity:

a. why did the authors not further define the profile of macrophages in the tissues, even if only by transcriptomics?

R. Following the important suggestions made by the reviewer, we recovered data from the RNAseq to identify genes related to M1/M2 polarization (Fig 3d) and we also analyzed the M2 macrophages subpopulation (Arg1+) in the lungs using IF (Fig 3c). To increase the significance of these analyses, we increased the number of samples, including both sexes. Histopathological data obtained from these lungs were added as well, in order to better characterize the lung pathology (Fig 3a). All these data are commented in the results and discussion sections (pages 8-9). The new Fig 3 now extensively recapitulates all these findings and should provide a clear identification of macrophages present in the lung of SARS-CoV-2-infected hamsters and the characterization of their transcriptomic profile related to polarization.

b. $\alpha 7$ nAChR activation increases HLA-DR, CD11b and CD54 expression but decreases CD14 (Siniavin, 2020) - how do the authors link to such findings?

R. We did not find changes in the gene expression of CD14 in the lungs of IVM-treated hamsters in comparison with non-treated ones (males and females). CD54 (Icam1, ENSMAUG00000006369) is slightly upregulated in both sexes and CD11b is slightly upregulated in females (ENSMAUG00000014830) (Dataset EV1). Based on our data, we cannot state that these markers are involved in the IVM's action in SARS-CoV-2-infected hamsters. Finally, HLA-DR is not described in the golden hamster. Nevertheless, by means of RNAseq and IF (new Fig 3), we raise the possibility that IVM modulates the expression of other surface markers in macrophages, including Arg1, an important M2-polarization marker. This is now discussed on page 9 (lines 7-14).

5. Clinical manifestation are linked to immune mediated dysregulation of coagulation - any observations here? There were some actions on platelet pathways; how do the authors link that to possible beneficial outcomes?

R. The IVM treatment acts on the coagulation pathway (Fig 2a), controlling the activation of platelets. As we stated in page 6 (lines 20-21), this action would be beneficial to prevent thrombosis, which has been considered the most important cause of death in COVID-19 patients (Chen & Pan, 2021). By histopathology (new Fig 3b) we could observe that circulatory alterations (thrombosis, congestion, microhemorrhages) were less intense in the lungs of IVM-treated animals in comparison to saline-treated ones. This is now presented and discussed on page 8 (lines 19-27).

6. What is the possible connection to nAChR here? It is mentioned several times but not expanded upon. It is known that activating $\alpha 7$ nAChR inhibits NLRP3 through β -arrestin-1 but no effect on IL-1 β was observed. Also, is there a differential expression profile of this receptor in immune cells? There are reported sex differences in $\beta 2$ nAChR expression could that be true for $\alpha 7$ also?

R. As we stated before (point #3), our RNAseq analyses did not show significant differences in the gene expression of NLRP3 nor IL-1 β in the lungs of IVM-treated hamsters. Conversely, we could identify that the "Cholinergic synapse" pathway was impacted in IVM-treated animals (Fig 2a). Indeed, we found that IVM-treated hamsters (males and females) presented a decreased gene expression of the CHRNA4, but, in our hands, IVM treatment did not affect CHRNA7 gene expression. Nevertheless, the increase or reduction of receptors gene expression may not be directly linked to their actions. In LPS-induced acute lung injury, it has been reported that a selective $\alpha 7$ nAChR agonist

(namely PNU-282987) reduced the concentrations of IL-6, TNF- α , and IL-1 β (Zhao et al., 2019), similarly to what we observed in the lungs of SARS-CoV-2-infected and IVM-treated hamsters (Fig 2b, Fig EV5b).

Of note, we have engaged new experiments to try to further understand the mechanisms of action of IVM at the molecular level, especially those related to the α 7nAChR. In ongoing experiments, SARS-CoV-2-infected and IVM-treated hamsters were injected with known antagonists of the nAChRs, and remarkably, these compounds blocked the beneficial effects of IVM treatment (clinical score and olfaction performance). Of course, these preliminary observations will be completed by using a variety of nAChRs orthosteric and allosteric ligands and by analyzing the immune profile in addition to the clinical aspects. These data will be presented in a separate publication.

Referee #3 (Comments on Novelty/Model System for Author):

Technical quality of the study is rated medium as more in-depth analyses would strengthen the manuscript (e.g. more detailed Analysis of cell recruitment to the respiratory tract). Novelty is high as treatments for COVID-19 are lacking and the study tests a novel candidate to modulate inflammation in COVID-19. Medical Impact is rated medium, as it is not clear how well it translates from the hamster model to the patient's situation. Nevertheless, the model system is adequate, as permissive animal models are an important tool for analyses in vivo.

R. We thank the reviewer for his comments and would like to underline that a more detailed analysis of cell recruitment and in particular of myeloid cells is now provided in this revised version.

Referee #3 (Remarks for Author):

In their manuscript "Attenuation of clinical and immunological outcomes during SARS-CoV-2 infection by ivermectin" the authors analyze immune modulatory effects of ivermectin on SARS-CoV-2 induced disease in an experimental hamster model. Due to the immunological component in human COVID-19, the lack in effective treatments and the fact that ivermectin is an approved drug in humans, this is certainly an interesting and potentially relevant study. Overall, the manuscript is well written and the data are clearly presented. Nevertheless, there are some issues to be addressed to improve clarity and to strengthen the proposed therapeutic potential of ivermectin.

1) Introduction p 3, line 15:

The authors should provide more detailed information on how ivermectin modulates the host's immune system. Also it should be indicated whether these data are derived from human or animal studies. It does not become especially clear why the authors hypothesize an anti-parasitic drug to affect SARS-CoV-2 infection in a hamster model.

R. We added more information about ivermectin's immunomodulation in page 3 (lines 13-16), nevertheless, its exact mechanism of action is not yet fully understood. Of note, the available data were provided from both human and animals studies.

The first data about the potential activity of ivermectin on SARS-CoV-2 was published by Caly and colleagues in 2020 (Caly et al., 2020), and it is also based on the ivermectin's potential wide range of action, specially over a multitude of viruses, including Dengue virus, Newcastle virus, Avian influenza A virus, HIV-1, including other coronaviruses (Arévalo et al., 2021, Heidary & Gharebaghi, 2020, Laing et al., 2017). These data are now also added in page 3 (lines 18-22).

2) p 4, line 24:

Are the data referred to presented in Fig EV1 or rather Fig 1b? A p-value of 0.018 (Fisher's exact test) is stated in the text and a p-value of 0.017 (Mantel Cox) is stated in Fig 1b for the respective comparison. Please clarify how/why these statistic tests were performed.

"The olfactory performance was also influenced by sex: 83.3% (10/12) of the saline-treated infected males presented hyposmia/anosmia, against only 33.3% (4/12) of IVM-treated infected males (Fisher's exact test p=0.036). In contrast, no olfactory deficit was observed in IVM-treated infected females (0/6), while 33.3% (2/6) of saline treated infected females presented with hyposmia/anosmia (Fisher's exact test p=0.455)." When stating that olfactory performance is influenced by sex, shouldn't males and females (uninfected as well as infected) be compared directly and a p-value provided? Has olfaction been statistically analysed between uninfected and IVM-treated infected females? Looking at figure EV1b it seems that even though all IVM-treated infected females eventually find the food, they take longer to do so than the uninfected controls. Therefore, is it feasible to state that no olfactory deficit was observed in IVM-treated females (p4, line 26)?

R. The data concerning olfactory tests were analyzed using both Mantel-Cox (to compare curves) and Fisher's test (to compare the final percentages). To clarify these comparisons, we added a new contingency graph dedicated to Fisher's test, where we have stated all the p-values (Fig EV1d).

3) p 7, line 23

"IVM limits the expression of Ifn-b, Ifn-l and of the IFN stimulated genes Cxcl10 and Mx2 in the lung of infected hamsters compared to infected and saline-treated hamsters."

As data are shown for females and males separately, the authors should indicate whether this statement refers to one or both sexes (as e.g. IFN-lambda is not affected by IVM in males).

R. We now indicate the differences by sex in page 7 (lines 23-26).

4) p4, line 24

"In contrast, saline treated hamsters presented an increased expression of Ifn-l, Cxcl10 and Mx2 compared to noninfected hamsters."

Please specify if this refers to males, females or both. For CXCL-10 also IVM treated hamsters still show about 4-fold higher transcription over uninfected hamsters. This should therefore be rewritten not to imply that expression is reduced to levels observed in uninfected hamsters.

R. We now indicate the differences by sex in page 7 (lines 23-26). To be stricter regarding gene expression, we rewrote the sentence by removing CXCL10.

5) p7, line 26

Regarding type I and III IFN, their induction following infection and IVM-mediated effects, the

authors should discuss the differences observed between the nasal turbinates and the lung and what this possibly implies for the observed overall host response/inflammation/clinical score. Generally, transcription of cytokine genes and effects of IVM in many points differ between the nasal turbinates and the lung and this should be discussed.

R. Following the reviewer's suggestions, we discussed the differences between nasal turbinates and lungs regarding type I and III IFNs in page 7 (lines 29-33) and page 8 (lines 1-2).

6) p7, line 34

Please also provide the IL-6/IL-10 ratio for the nasal turbinates. Is this a consistent finding between the lung and nasal turbinates?

R. We have now added the IL-6/IL-10 ratio for the nasal turbinates in Fig 1f. There was no difference for IL-6/IL-10 ratio in the nasal turbinates in IVM-treated or saline-treated animals. We have also included this information in page 8 (line 11).

7) p8, line 5

Recruitment of leucocytes to the lungs is an important aspect of the modulation of inflammation by IVM and the data shown are very interesting. However, more data should be presented to underline the findings. Only 2 mice were analyzed for the treatment group. For example, analysis of H&E stained lungs, differential cell counts of bronchoalveolar lavage or flow cytometric analyses could provide more insight into cell recruitment to the lungs in infection, effects of IVM and correlations to the cytokine/chemokine profile.

R. During the setting up of the hamster model, we tried indeed to perform bronchoalveolar lavage, but due to several technical issues (manipulation in BSL-3 isolators) and anatomical issues (small size of animals and organs) we abandoned this idea.

Since the initial submission, we repeated these experiments to add more animals, and the present version of the manuscript contains 6 animals (Fig 3c). In this version we also added histopathological data obtained from these lungs, in order to better characterize the lung pathology (Fig 3a). To further characterize these myeloid cells, we recovered data from the RNAseq to identify genes related to M1/M2 polarization (Fig 3d) and we also analyzed the M2 polarization of these cells in the lungs using IF and the Arg1 antibody (Fig 3b). These data are now presented and discussed in pages 8-9. This new Fig3 now extensively recapitulates all these findings and should provide a clear identification of macrophages present in the lung of SARS-CoV-2-infected hamsters and the characterization of their transcriptomic profile related to polarization

8) Regarding the timing of treatment in the model, what is the rationale for treating the hamsters together with the infection? Do the authors have any data on a more therapeutic potential of ivermectin, i.e. its effects when treating hamsters e.g. on day 2 or day 4 post infection when viral replication and infection have established?

R. When studying the therapeutic effects of a molecule *in vivo*, the first approach is indeed the administration of the treatment at the time of infection, or even before the infection. This approach was indeed used by others in the context of COVID-19 (Driouich et al., 2021, Kaptein et al., 2020).

The golden hamster is now a well-established model for moderate-to-severe COVID-19. The infected hamsters do not die from the infection, and the infectious process will be solved after 5-6 days post-infection, when animals start to gain weight and viral titers start to decrease (Chan et al., 2020, de Melo et al., 2021, Sia et al., 2020). After 2- or 4-days post-infection it would not be possible to distinguish from the effects of a therapy or the effects of the self-healing process observed in this model. As a matter of fact, we have chosen the time-point of 4 days post-infection to euthanize animals to perform all the analyses included in this manuscript based on the published data and on our own observations (see Figure A below, obtained from one of our recently published manuscripts: (de Melo et al., 2021).

Figure A. Clinical and molecular characteristics of experimental infection with SARS-CoV-2 in golden hamsters.

We can consider that the infectious process in golden hamsters is established far before 1-day post-infection, when the viral titers has already increased by 5-logs (see Figure B below, personal communication). In hamsters, the SARS-CoV-2 infectious process does not develop at the same speed as that observed in humans, it is much faster; therefore, as commonly performed in *in vivo* drug testing, we need to perform an early treatment to have enough time to observe some effects.

Figure B. Viral titers in the nasal turbinates and in the lungs of SARS-CoV-2-infected golden hamsters.

9) The authors should provide more detailed information on the hamster model. Is it a lethal or sublethal model and is the virus eventually cleared. Here, hamsters were sacrificed 4 days post

infection for analysis. To further assess the therapeutic potential, it would be important to assess whether IVM is able to improve survival. Along these lines, it is particularly stunning that IVM treated hamsters (especially females) show such little clinical scores until day 4 despite unchanged viral load in the respiratory tract. Also therefore, survival studies and a later time-point of analysis would need to confirm the potential of IVM. Possibly, in the later course the benefit of IVM treatment is overcome by a continuously high viral load.

R. As we stated in the answer to the point 8, the infected hamsters do not die from the infection (Chan et al., 2020, de Melo et al., 2021, Sia et al., 2020), regardless of the infectious dose (de Melo et al., 2021; Imai et al., 2020), therefore this model cannot be used to analyze survival rates. However, this model is robust enough to analyze therapeutic or even prophylactic approaches based on the clinical presentation of the animals. We added more information about the hamster model in page 4 (lines 14-19) and the reader may find more information about the model in de Melo et al. (2021) which recently came out.

10) Overall, from the data presented and discussed it does not become clear how the authors believe IVM acts upstream of the inflammatory response and how the strong difference between males and females regarding the effects of IVM treatment are mediated. Possibly the authors could elaborate more on these points.

R. We have further reinforced and documented our comments and discussion regarding the sex-related effects that we observed in hamsters throughout the text, and in page 10 (lines 11-18). Regarding upstream action of IVM, IVM might possibly modulate *several* upstream targets or biomarkers of inflammation (e.g. cytokine gene expression, cell activation), rather than the inhibition of a *single* downstream mediator which would have more limited effects. We further documented our hypothesis in page 10 (lines 1-8).

References

- Arévalo AP, Pagotto R, Pórfido JL, Daghero H, Segovia M, Yamasaki K, Varela B, Hill M, Verdes JM, Duhalde Vega M et al. (2021) Ivermectin reduces in vivo coronavirus infection in a mouse experimental model. *Sci Rep* 11: 7132
- Bunders MJ, Altfeld M (2020) Implications of Sex Differences in Immunity for SARS-CoV-2 Pathogenesis and Design of Therapeutic Interventions. *Immunity* 53: 487-495
- Caly L, Druce JD, Catton MG, Jans DA, Wagstaff KM (2020) The FDA-approved drug ivermectin inhibits the replication of SARS-CoV-2 in vitro. *Antiviral Res* 178: 104787
- Cerdan AH, Sisqueiras M, Pereira G, Barreto Gomes DE, Changeux JP, Cecchini M (2020) The Glycine Receptor Allosteric Ligands Library (GRALL). *Bioinformatics* 36: 3379-3384
- Chan JF, Zhang AJ, Yuan S, Poon VK, Chan CC, Lee AC, Chan WM, Fan Z, Tsoi HW, Wen L et al. (2020) Simulation of the clinical and pathological manifestations of Coronavirus Disease 2019 (COVID-19) in golden Syrian hamster model: implications for disease pathogenesis and transmissibility. *Clin Infect Dis*
- Chen W, Pan JY (2021) Anatomical and Pathological Observation and Analysis of SARS and COVID-19: Microthrombosis Is the Main Cause of Death. *Biological Procedures Online* 23: 4

Cross SJ, Linker KE, Leslie FM (2017) Sex-dependent effects of nicotine on the developing brain. *Journal of Neuroscience Research* 95: 422-436

de Melo GD, Lazarini F, Levallois S, Hautefort C, Michel V, Larrous F, Verillaud B, Aparicio C, Wagner S, Gheusi G et al. (2021) COVID-19-related anosmia is associated with viral persistence and inflammation in human olfactory epithelium and brain infection in hamsters. *Science Translational Medicine* doi: 10.1126/scitranslmed.abf8396

Driouich J-S, Cochin M, Lingas G, Moureau G, Touret F, Petit P-R, Piorkowski G, Barthélémy K, Laprie C, Coutard B et al. (2021) Favipiravir antiviral efficacy against SARS-CoV-2 in a hamster model. *Nature Commun* 12: 1735

González Canga A, Sahagún Prieto AM, Díez Liébana MJ, Fernández Martínez N, Sierra Vega M, García Vieitez JJ (2008) The Pharmacokinetics and Interactions of Ivermectin in Humans—A Mini-review. *The AAPS Journal* 10: 42-46

González Canga A, Sahagún Prieto AM, José Díez Liébana M, Martínez NF, Vega MS, Vieitez JJG (2009) The pharmacokinetics and metabolism of ivermectin in domestic animal species. *The Veterinary Journal* 179: 25-37

Guzzo CA, Furtek CI, Porras AG, Chen C, Tipping R, Clineschmidt CM, Sciberras DG, Hsieh JY, Lasseter KC (2002) Safety, tolerability, and pharmacokinetics of escalating high doses of ivermectin in healthy adult subjects. *Journal of clinical pharmacology* 42: 1122-33

Heidary F, Gharebaghi R (2020) Ivermectin: a systematic review from antiviral effects to COVID-19 complementary regimen. *The Journal of Antibiotics* 73: 593-602

Imai M, Iwatsuki-Horimoto K, Hatta M, Loeber S, Halfmann PJ, Nakajima N, Watanabe T, Ujje M, Takahashi K, Ito M et al. (2020) Syrian hamsters as a small animal model for SARS-CoV-2 infection and countermeasure development. *Proc Natl Acad Sci U S A* 117: 16587-16595

Kaptein SJF, Jacobs S, Langendries L, Seldeslachts L, ter Horst S, Liesenborghs L, Hens B, Vergote V, Heylen E, Barthelemy K et al. (2020) Favipiravir at high doses has potent antiviral activity in SARS-CoV-2-infected hamsters, whereas hydroxychloroquine lacks activity. *Proceedings of the National Academy of Sciences* 117: 26955-26965

Laing R, Gillan V, Devaney E (2017) Ivermectin – Old Drug, New Tricks? *Trends in Parasitology* 33: 463-472

Lifschitz A, Virkel G, Sallovitz J, Sutra JF, Galtier P, Alvinerie M, Lanusse C (2000) Comparative distribution of ivermectin and doramectin to parasite location tissues in cattle. *Veterinary Parasitology* 87: 327-338

Scully EP, Haverfield J, Ursin RL, Tannenbaum C, Klein SL (2020) Considering how biological sex impacts immune responses and COVID-19 outcomes. *Nature Reviews Immunology* 20: 442-447

Sia SF, Yan LM, Chin AWH, Fung K, Choy KT, Wong AYL, Kaewpreedee P, Perera R, Poon LLM, Nicholls JM et al. (2020) Pathogenesis and transmission of SARS-CoV-2 in golden hamsters. *Nature* 583: 834-838

Takahashi T, Ellingson MK, Wong P, Israelow B, Lucas C, Klein J, Silva J, Mao T, Oh JE, Tokuyama M et al. (2020) Sex differences in immune responses that underlie COVID-19 disease outcomes. *Nature* 588: 315-320

vom Steeg LG, Klein SL (2019) Sex and sex steroids impact influenza pathogenesis across the life course. *Seminars in Immunopathology* 41: 189-194

Zhao X, Yu Z, Lv Z, Meng L, Xu J, Yuan S, Fu Z (2019) Activation of Alpha-7 Nicotinic Acetylcholine Receptors ($\alpha 7nAChR$) Promotes the Protective Autophagy in LPS-Induced Acute Lung Injury (ALI) In Vitro and In Vivo. *Inflammation* 42: 2236-2245

18th Jun 2021

Dear Dr. Bourhy,

Thank you for the submission of your manuscript to EMBO Molecular Medicine. I am pleased to inform you that we will be able to accept your manuscript pending the following final amendments:

- 1) Please address all referee #3 comments.
- 2) Figures: We noticed that Figure 1B and EV Figures 1B and 2C contain data from the same experiment (CoV_saline and CoV_ivermectin). Please clarify in the figure legends.
- 3) Tables: Please move Table 1 and 2 to the Appendix and rename them to "Appendix Table S1 and S2". Also, correct their callouts in the text accordingly.
- 4) In the main manuscript file, please do the following:
 - Correct/answer the track changes suggested by our data editors by working from the attached document.
 - Remove red font colour.
 - Make sure that all special characters display well.
 - In M&M, a statistical paragraph should reflect all information that you have filled in the Authors Checklist, especially regarding randomization, blinding, replication.
 - In addition to the accession number please provide URL for RNA Seq data. Using accession number E-MTAB-10128 we could not access the data. Please be aware that all datasets should be made freely available upon acceptance, without restriction. Use the following format to report the accession number of your data:

[data type]: [full name of the resource] [accession number/identifier] ([doi or URL or identifiers.org/DATABASE:ACCESSION])

Please check "Author Guidelines" for more information.

<https://www.embopress.org/page/journal/17574684/authorguide#availabilityofpublishedmaterial>

- 5) Appendix: Please add table of content and figure legend.
- 6) Synopsis: Please check your synopsis text and image, revise them if necessary and submit their final versions with your revised manuscript. Please be aware that in the proof stage minor corrections only are allowed (e.g., typos).
- 7) For more information: There is space at the end of each article to list relevant web links for further consultation by our readers. Could you identify some relevant ones and provide such information as well? Some examples are patient associations, relevant databases, OMIM/proteins/genes links, author's websites, etc...
- 8) Source data: We encourage you to include the source data for figure panels that show essential data. Numerical data should be provided as individual .xls or .csv files (including a tab describing the data). For blots or microscopy, uncropped images should be submitted (using a zip archive if multiple images need to be supplied for one panel). Please check "Author Guidelines" for more information. <https://www.embopress.org/page/journal/17574684/authorguide#sourcedata>
- 9) Press release: Please inform us as soon as possible and latest at the time of submission of the revised manuscript if you plan a press release for your article so that our publisher could coordinate publication accordingly.
- 10) Please be aware that we use a unique publishing workflow for COVID-19 papers: a non-typeset PDF of the accepted manuscript is published as "Just Accepted" on our website. With respect to a

possible press release, we have the option to not post the "Just Accepted" version if you prefer to wait with the press release for the typeset version. Please let us know whether you agree to publication of a "Just accepted" version or you prefer to wait for the typeset version.

11) As part of the EMBO Publications transparent editorial process initiative (see our Editorial at <http://embomolmed.embopress.org/content/2/9/329>), EMBO Molecular Medicine will publish online a Review Process File (RPF) to accompany accepted manuscripts. This file will be published in conjunction with your paper and will include the anonymous referee reports, your point-by-point response and all pertinent correspondence relating to the manuscript. Let us know whether you agree with the publication of the RPF and as here, if you want to remove or not any figures from it prior to publication. Please note that the Authors checklist will be published at the end of the RPF.

12) Please provide a point-by-point letter INCLUDING my comments as well as the reviewer's reports and your detailed responses (as Word file).

I look forward to reading a new revised version of your manuscript as soon as possible.

Yours sincerely,

Zeljko Durdevic

***** Reviewer's comments *****

Referee #1 (Comments on Novelty/Model System for Author):

This is a very relevant topic as the question of ivermectin and COVID-19 has already led to policy in several countries around the world. This information will support the conduction of appropriately powered clinical trials.

Referee #1 (Remarks for Author):

I am satisfied with the way my comments have been answered/addressed.

Referee #2 (Comments on Novelty/Model System for Author):

Already commented on first review

Referee #2 (Remarks for Author):

I am satisfied with the authors response to the review, and the changes to the manuscript

Referee #3 (Remarks for Author):

The authors have sufficiently addressed the comments made on their originally submitted version of the manuscript. New data have been added and a number of points have been clarified and discussed. Through the revision, the manuscript has improved substantially.

There are a few minor issues left that in my opinion need to be addressed:

- 1) p4, line 10 - "...exhibited the strongest clinical outcome." Is this correct? It implies treated females exhibited the most signs of disease.
- 2) p8, line 10 - for significant overexpression of IL-10 authors refer to Fig 2b and EV4. It is not clear, where in EV4 this information is provided.
- 3) p9, lines 11 to 14 - check whether Fig 3c is correct/correctly labelled as description in the text does not seem to fully fit. E.g. is the left panel Iba1+Arg-? In the right panel (labelled Iba1+Arg2+) it is hard to make out an increase in Iba1+Arg1+ cells in the lungs of treated males, while this is depicted in the middle panel (labelled Arg1+).

The authors performed the requested editorial changes.

***** Reviewer's comments *****

Referee #1 (Comments on Novelty/Model System for Author):

This is a very relevant topic as the question of ivermectin and COVID-19 has already led to policy in several countries around the world. This information will support the conduction of appropriately powered clinical trials.

Referee #1 (Remarks for Author):

I am satisfied with the way my comments have been answered/addressed.

R. We wish to thank the reviewer for this positive comment.

Referee #2 (Comments on Novelty/Model System for Author):

Already commented on first review

Referee #2 (Remarks for Author):

I am satisfied with the authors response to the review, and the changes to the manuscript R.

We wish to thank the reviewer for this positive comment.

Referee #3 (Remarks for Author):

The authors have sufficiently addressed the comments made on their originally submitted version of the manuscript. New data have been added and a number of points have been clarified and discussed. Through the revision, the manuscript has improved substantially.

R. We thank the reviewer for these positive comments. Below are our answers to the different points raised.

There are a few minor issues left that in my opinion need to be addressed:

1) p4, line 10 - "...exhibited the strongest clinical outcome." Is this correct? It implies treated females exhibited the most signs of disease.

R. Actually females exhibited better clinical outcomes. We corrected this sentence.

2) p8, line 10 - for significant overexpression of IL-10 authors refer to Fig 2b and EV4. It is not clear, where in EV4 this information is provided.

R. IL-10 expression is showed only in Fig 2b. We deleted the caption for Fig EV4.

3) p9, lines 11 to 14 - check whether Fig 3c is correct/correctly labelled as description in the text does not seem to fully fit. E.g. is the left panel Iba1+Arg-? In the right panel (labelled Iba1+Arg2+) it is hard to make out an increase in Iba1+Arg1+ cells in the lungs of treated males, while this is depicted in the middle panel (labelled Arg1+).

R. We agreed that this point was a bit confusing. We rewrote this sentence (page 9, line 9) to better reflect the data in Figure 3c: "an increase of both Iba1+ and Arg1+ cells in the lungs of SARS-CoV-2-infected male hamsters (Fig 3c)".

We are pleased to inform you that your manuscript is accepted for publication and is now being sent to our publisher to be included in the next available issue of EMBO Molecular Medicine.

YOU MUST COMPLETE ALL CELLS WITH A PINK BACKGROUND ↓
PLEASE NOTE THAT THIS CHECKLIST WILL BE PUBLISHED ALONGSIDE YOUR PAPER

Corresponding Author Name: Bourhy H.
Journal Submitted to: EMBO Molecular Medicine
Manuscript Number: EMM-2021-14122